# ACHIEVING APPROXIMATE SYMMETRY IS EXPONENTIALLY EASIER THAN EXACT SYMMETRY

**Behrooz Tahmasebi & Melanie Weber**
Harvard John A. Paulson School of Engineering and Applied Sciences (SEAS)
Harvard University, Cambridge, MA 02138, USA
`{behrooz_tahmasebi,mweber}@seas.harvard.edu`

## ABSTRACT

Enforcing *exact symmetry* in machine learning models often yields significant gains in scientific applications, serving as a powerful inductive bias. However, recent work suggests that relying on *approximate symmetry* can offer greater flexibility and robustness. Despite promising empirical evidence, there has been little theoretical understanding, and in particular, a direct comparison between exact and approximate symmetry is missing from the literature. In this paper, we initiate this study by asking: *What is the cost of enforcing exact versus approximate symmetry?* To address this question, we introduce *averaging complexity*, a framework for quantifying the cost of enforcing symmetry via averaging. Our main result is an exponential separation: under standard conditions, exact symmetry requires linear averaging complexity, whereas approximate symmetry can be attained with only logarithmic complexity in the group size. To the best of our knowledge, this provides the first theoretical separation of these two cases, formally justifying why approximate symmetry may be preferable in practice. Beyond this, our tools and techniques may be of independent interest for the broader study of symmetries in machine learning.

## 1 INTRODUCTION

The field of *geometric machine learning* aims to incorporate *structures* observed in scientific data into abstract machine learning models, with the goal of leveraging these strong inductive biases to make learning more robust, efficient, and interpretable (Bronstein et al., 2021; Weber, 2025). Prominent examples include permutation symmetries in point clouds for vision tasks, sign-flip symmetries in spectral graph methods, rotational symmetry in robotic tasks, and other structures in molecular and atomistic data with applications from physics to drug discovery (Bogatskiy et al., 2020; Wang et al., 2022a; Nguyen et al., 2024; Kufel et al., 2025).

A natural approach to handling symmetries is to encode them *exactly* into the model through different mechanisms. This ensures that the invariance hypothesis is exploited to its full extent. The literature offers a variety of such methods, including model-agnostic approaches such as group averaging, data augmentation (Dao et al., 2019; Chen et al., 2020; Patil & Du, 2023; Tahmasebi et al., 2025), canonicalization, and frame averaging (Puny et al., 2022; Lin et al., 2024; Atzmon et al., 2022; Kaba et al., 2023; Ma et al., 2024; Tahmasebi & Jegelka, 2025a;b; Dym et al., 2024; Shumaylov et al., 2025), as well as model-dependent approaches such as convolutional neural networks and neural networks with equivariant weights (Cohen & Welling, 2016; 2017; Krizhevsky et al., 2012; Satorras et al., 2021; Maron et al., 2019; Liao & Smidt, 2023; Zaheer et al., 2017). Both categories have been shown to be effective in practice, and detailed theoretical studies have further analyzed their benefits.

However, introducing *exact symmetries* also comes with a number of caveats. In many applications, invariance is only partial, and targets may respect symmetry only approximately (Finzi et al., 2021; Romero & Lohit, 2022; van der Ouderaa et al., 2022; Kim et al., 2023; Park et al., 2025; Wang et al., 2022b). For example, in medical imaging, expected reflectional symmetries are not perfect, and results are often mildly sensitive to such transformations. Another case arises when only partial knowledge of the underlying symmetries is available, and symmetry discovery is performed (Yang

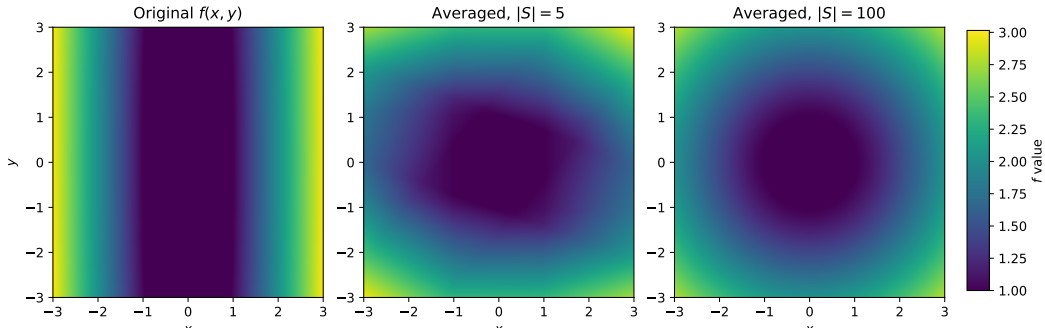

Figure 1: Approximate and exact symmetry enforcement via averaging for the 100-element group of 2D rotations. Left: original anisotropic function $f(x, y)$. Middle: average over $|S| = 5 \approx \log(100)$ random rotations (approximate symmetry). Right: average over $|S| = 100$ rotations (exact symmetry). Approximate symmetry is already high quality when $|S| \approx \log|G|$.

et al., 2023; van der Ouderaa et al., 2023; Desai et al., 2022; Dehmamy et al., 2021; Shaw et al., 2024; Yang et al., 2024; Huh, 2025). In this setting, enforcing exact invariance introduces fundamental limitations on universality and expressive power, making flexibility essential. Indeed, from a distributional shift, robustness, and optimization perspective, it is often argued that allowing the model to violate symmetry up to a certain degree can improve performance while still exploiting the strong inductive biases present in the data. Motivated by these considerations, researchers have proposed using *approximate symmetry* instead of exact symmetry, which enables models to be more flexible, to achieve more robust performance, and to exploit symmetry in a semi-supervised fashion, particularly in the context of symmetry discovery.

Despite these practical successes, theoretical gaps remain. Since approximate symmetry can be viewed as a relaxed form of invariance compared to hard-coded constraints, one might expect it to be *easier* to achieve in data. A theoretical analysis of the complexity of this emergence would provide several benefits. First, it explains why approximate symmetries are ubiquitous in data, where exact equivariance rarely holds. Second, it shows why models exploit them more easily: lower enforcement complexity can yield better sample or computational efficiency, as well as robustness to noise and distributional shifts.

Motivated by these considerations, in this paper, we study the following question: *Is it easier, from a complexity perspective, to enforce approximate symmetry compared to exact symmetry?* A key challenge lies in defining what is meant by the "complexity" of achieving approximate symmetry and in formalizing the associated "budget" in this setting. While there is no unified notion of such complexity in the literature, we introduce a natural measure for comparing the two regimes: *averaging complexity*.

To define averaging complexity, we assume access to a black-box model, and a learner that is only allowed to post-process this model linearly via a number of action queries (AQ). The number of such queries required in an averaging scheme is defined as the averaging complexity of the scheme. The learner's goal is to accomplish her task using as few queries as possible, which we interpret as the learner's budget.

Within this formal framework, we pose the following quantitative question: Given a model, what is the averaging complexity of enforcing approximate versus exact symmetry? Is there a separation between their complexities? Specifically, is achieving approximate symmetry easier than exact symmetry, as suggested by practical evidence?

Our main contribution is summarized in the following statement (informal; under mild conditions):

> The averaging complexity of achieving exact symmetry scales linearly with the group size, whereas approximate symmetry requires only logarithmic complexity.

This result provides a foundation for understanding why approximate symmetry is often preferred in practice: in an abstract model, it is *exponentially* easier to achieve. The central message of this paper is the exponential separation, demonstrating that for a given budget, approximate symmetry is more capable of achieving stronger results in semi-supervised learning (for example, in symmetry discovery). Beyond this, our abstract framework and complexity notion, together with the representation-theoretic tools developed in this work, can also be applied to the broader study of geometric machine learning, independent of the specific results presented here.

In short, this paper makes the following contributions:

- We advance the *theoretical* understanding of approximate versus exact symmetries in machine learning models, and we prove that approximate symmetry is exponentially easier to enforce in an abstract setting. To the best of our knowledge, this is the first theoretical separation between these two widely used approaches.

- The abstract formulation of averaging complexity, together with the theoretical tools developed in this work, may be of independent interest for future studies in the theory of geometric machine learning. We believe that the results presented here represent just one instance of their broader applicability.

## 2 RELATED WORK

Symmetries appear in many scientific datasets, and equivariant machine learning has proven powerful across applications in particle physics (Bogatskiy et al., 2020), robotics (Wang et al., 2022a), and quantum physics, in both exact (Nguyen et al., 2024) and approximate forms (Kufel et al., 2025). Incorporating symmetry has been shown to improve sample complexity and generalization (Wang et al., 2021; Tahmasebi & Jegelka, 2023; Elesedy, 2021), estimation (Chen et al., 2023; Tahmasebi & Jegelka, 2024), and learning complexity (Kiani et al., 2024; Soleymani et al., 2025c;a). Generalization benefits have been observed even when only approximate symmetry holds (Petrache & Trivedi, 2023).

Many architectures have been proposed for incorporating symmetries in neural networks, including group-equivariant CNNs (Cohen & Welling, 2016) and steerable CNNs (Cohen & Welling, 2017), both built on top of standard convolutional networks (Krizhevsky et al., 2012). Equivariant graph neural networks (Satorras et al., 2021; Maron et al., 2019) and transformers (Liao & Smidt, 2023) have also been proposed and used in practice. A canonical example for permutation symmetry is Deep Sets (Zaheer et al., 2017).

Beyond exact methods, many approaches for introducing relaxed invariance have been proposed in the literature, including modified filters (van der Ouderaa et al., 2022), soft equivariance (Kim et al., 2023; Finzi et al., 2021), partial equivariance (Romero & Lohit, 2022), and Lie-algebraic parameterizations (McNeela, 2023). Approximate symmetry has proved effective in reinforcement learning (Park et al., 2025) via approximately equivariant Markov decision processes (MDPs). Other examples include the use of structured matrices (Samudre et al., 2025) and relaxed constraints (Pertigkiozoglou et al., 2024); see also (Wang et al., 2022b; Wu et al., 2025). For neural processes, Ashman et al. (2024) propose approximately equivariant schemes with promising benefits. This line of work extends to approximately equivariant graph networks (Huang et al., 2023) and symmetry breaking for relaxed equivariance (Wang et al., 2024; 2023). The role and benefits of approximate equivariance in the neural-network optimization landscape have also been studied (Xie & Smidt, 2025). In the context of symmetry discovery, many results use semi-supervised methods to learn the underlying symmetry (Yang et al., 2023; van der Ouderaa et al., 2023; Desai et al., 2022; Dehmamy et al., 2021; Shaw et al., 2024; Yang et al., 2024; Huh, 2025).

For model-agnostic methods for equivariant learning, see frame averaging (Puny et al., 2022; Lin et al., 2024; Atzmon et al., 2022) and canonicalization (Kaba et al., 2023; Ma et al., 2024; Tahmasebi & Jegelka, 2025a;b; Dym et al., 2024; Shumaylov et al., 2025) as two widely applicable paradigms.

## 3 PROBLEM STATEMENT

This section formulates the problem and introduces the function spaces and notation used throughout the paper. The main results appear in the next section, with detailed background provided in Appendix A.

### 3.1 PRELIMINARIES, NOTATION, AND BACKGROUND

Given $n \in \mathbb{N}$, we write $[n] := \{1, 2, \ldots, n\}$. Let $\mathcal{X}$ be a topological space (the data domain), and let $G$ be a finite group. Let $L^2(\mathcal{X})$ denote the space of square-integrable functions on $\mathcal{X}$, assuming $\mathcal{X}$ is equipped with a canonical Borel measure $\mu$.

A (left) *group action* of $G$ on $\mathcal{X}$ is a map $\theta : G \times \mathcal{X} \to \mathcal{X}$ such that $\theta(gh, x) = \theta\big(g, \theta(h, x)\big)$ for all $g, h \in G$ and $x \in \mathcal{X}$, and the identity element of $G$ acts trivially (via the identity map $x \mapsto x$) on $\mathcal{X}$. We write $gx := \theta(g, x)$; for each $g$, the map $x \mapsto gx$ is a homeomorphism of $\mathcal{X}$. Indeed, without loss of generality, we assume that the canonical measure $\mu$ on $\mathcal{X}$ is invariant under the action of $G$.

Let $\mathcal{F} \subseteq L^2(\mathcal{X})$ be a finite-dimensional real vector space of continuous functions on $\mathcal{X}$, and let $GL(\mathcal{F})$ denote the group of invertible linear mappings from $\mathcal{F}$ to itself (under composition). Assume that for every $g \in G$ and $f \in \mathcal{F}$, the function $x \mapsto f(gx)$ also belongs to $\mathcal{F}$. Define $\rho : G \to GL(\mathcal{F})$ as the canonical group action on $\mathcal{F}$ by leveraging the action on the domain:

$$(\rho(g)[f])(x) := f\left(g^{-1}x\right), \qquad \forall f \in \mathcal{F}, \ \forall x \in \mathcal{X}.$$

Indeed, $\rho$ is a (linear) group representation of $G$ on $\mathcal{F}$, meaning that $\rho(gh) = \rho(g)\rho(h)$ under the composition of linear maps.

*Remark* 1. While the results in this paper are mainly framed as achieving *invariance* via averaging, they all follow using the same procedure to achieve *equivariance* via averaging. Using a natural algebraic correspondence, one can find a bijection between such equivariant functions and invariant functions on a new appropriate space. We detail this construction in Appendix A.4.

*Remark* 2. The results of this paper extend verbatim to finite-dimensional complex subspaces of $L^2(\mathcal{X})$, without requiring real-valuedness or continuity. We restrict attention to real-valued continuous functions for notational and conceptual simplicity.

### 3.2 AVERAGING SCHEMES

In this part, we formalize *averaging schemes* as abstract mechanisms for enforcing desired functional properties (e.g., symmetry) in function classes.

Consider an abstract setting where a learner aims to post-process the function class $\mathcal{F}$ to enforce a condition (e.g., symmetry). The learner is informed that an arbitrary function $f \in \mathcal{F}$ has been chosen by an oracle and that it remains unchanged throughout post-processing. The learner then issues functional queries to the oracle as follows. Given $f \in \mathcal{F}$ and a group element $g \in G$, the oracle returns the transformed function $x \mapsto f(gx) \in \mathcal{F}$. Because each query evaluates $f$ on $gx \in \mathcal{X}$, we call it an *action query (AQ)*.

After issuing a number of action queries, the learner forms a linear combination of the oracle responses to obtain a post-processed function. The learner has a limited budget and seeks to minimize the number of action queries. This motivates the following definition.

**Definition 3** (Averaging Scheme). *An averaging scheme is a function* $\omega : G \to \mathbb{R}$ *on the finite group* $G$ *such that* $\|\omega\|_{\ell_1(G)} := \sum_{g \in G} \omega(g) = 1$. *For a function class* $\mathcal{F}$, *the averaging operator induced by* $\omega$, *denoted* $\mathbb{E}_\omega : \mathcal{F} \to \mathcal{F}$, *is defined by*

$$(\mathbb{E}_\omega[f])(x) := \sum_{g \in G} \omega(g) f\left(g^{-1}x\right), \qquad \forall f \in \mathcal{F}, \ \forall x \in \mathcal{X}.$$

*The* size *of an averaging scheme is the number of nonzero weights:*

$$\mathrm{size}(\omega) := \#\{ g \in G : \omega(g) \neq 0 \}.$$

Intuitively, an averaging scheme specifies weights used to linearly combine the transformed functions $x \mapsto f\left(g^{-1}x\right)$ to produce the final output. Crucially, averaging schemes *do not* depend on

the domain point $x \in \mathcal{X}$; otherwise, they become instances of (weighted) frame averaging, and the notion of averaging complexity becomes ill-defined. We therefore focus on *universal* linear combinations as outputs of averaging operators.

*Remark* 4. One may allow $\omega$ to take complex values without affecting the results. For simplicity, however, we restrict attention to real-valued averaging schemes $\omega$.

### 3.3 AVERAGING COMPLEXITY

In this paper, we consider the abstract setting where the learner aims to obtain either an exactly symmetric function or an approximately symmetric one. To define *averaging complexity*, we first introduce a few definitions, starting with exact symmetry.

**Definition 5** (Exact Symmetry). *A function $f \in \mathcal{F}$ is* exactly symmetric *if, for all $g \in G$ and all $x \in \mathcal{X}$, one has $f(gx) = f(x)$.*

To define approximate symmetry, one must fix a notion of distance from symmetry and allow a relaxation within a prescribed precision. A natural choice is to shrink the "non-symmetry" components of functions (in $L^2(\mathcal{X})$) by a small factor $\epsilon > 0$. When $\epsilon = 0$, the definition reduces to exact symmetry. The $L^2(\mathcal{X})$-norm is a canonical way to define distances in function space, and this particular choice for defining different notions of approximate symmetry enables our application to the generalization theory of approximately symmetric regression; see Appendix A.8. For further discussion on going beyond the $L^2(\mathcal{X})$-norm, please see Appendix F.

In this paper, we use two types of approximate symmetry: *weak* and *strong*.

**Definition 6** (Weak Approximate Symmetry Enforcement). *An averaging scheme $\omega : G \to \mathbb{R}$ enforces weak approximate symmetry* with respect to a parameter $\epsilon > 0$ *if and only if, for every function $f \in \mathcal{F}$, we have*

$$\mathbb{E}_g \left[ \int_{\mathcal{X}} |(\mathbb{E}_\omega[f])(x) - (\mathbb{E}_\omega[f])(gx)|^2 \, d\mu(x) \right] \leq \epsilon \, \mathbb{E}_g \left[ \int_{\mathcal{X}} |f(x) - f(gx)|^2 \, d\mu(x) \right],$$

*where $g \in G$ is chosen uniformly at random and $\mu$ is the canonical Borel measure on $\mathcal{X}$.*

**Definition 7** (Strong Approximate Symmetry Enforcement). *An averaging scheme $\omega : G \to \mathbb{R}$ enforces strong approximate symmetry* with respect to a parameter $\epsilon > 0$ *if and only if, for every function $f \in \mathcal{F}$, we have*

$$\int_{\mathcal{X}} |(\mathbb{E}_\omega[f])(x) - (\mathbb{E}_\omega[f])(gx)|^2 \, d\mu(x) \leq \epsilon \, \mathbb{E}_g \left[ \int_{\mathcal{X}} |f(x) - f(gx)|^2 \, d\mu(x) \right], \qquad \forall g \in G,$$

*where $g \in G$ is chosen uniformly at random and $\mu$ is the canonical Borel measure on $\mathcal{X}$.*

In the weak notion, $\mathbb{E}_\omega[f]$ is multiplicatively $\epsilon$-closer (in $L^2(\mathcal{X})$) to being symmetric *on average* over group elements $g \in G$. In the strong notion, the same closeness must hold *for every* $g \in G$.

We are now ready to define the concept of averaging complexity.

**Definition 8** (Averaging Complexity). *The* averaging complexity *of enforcing exact, weak approximate, or strong approximate symmetry, denoted $\mathsf{AC}^{\mathrm{ex}}(\mathcal{F})$, $\mathsf{AC}^{\mathrm{wk}}(\mathcal{F}, \varepsilon)$, and $\mathsf{AC}^{\mathrm{st}}(\mathcal{F}, \varepsilon)$, respectively, is the minimal size of an averaging scheme that a learner can construct such that the resulting post-processed function is exactly, weakly approximately, or strongly approximately symmetric, respectively. Formally,*

$$\mathsf{AC}^{\mathrm{ex}}(\mathcal{F}) \quad := \min_\omega \, \mathrm{size}(\omega) \qquad s.t. \quad (\mathbb{E}_\omega[f])(gx) = (\mathbb{E}_\omega[f])(x), \quad \forall f \in \mathcal{F}, g \in G, x \in \mathcal{X}$$

$$\mathsf{AC}^{\mathrm{wk}}(\mathcal{F}, \varepsilon) := \min_\omega \, \mathrm{size}(\omega) \qquad s.t. \quad \mathbb{E}_g \left[ \|(\mathbb{E}_\omega[f])(x) - (\mathbb{E}_\omega[f])(gx)\|_{L^2(\mathcal{X})}^2 \right]$$
$$\leq \varepsilon \, \mathbb{E}_g \left[ \|f(x) - f(gx)\|_{L^2(\mathcal{X})}^2 \right], \quad \forall f \in \mathcal{F}$$

$$\mathsf{AC}^{\mathrm{st}}(\mathcal{F}, \varepsilon) \quad := \min_\omega \, \mathrm{size}(\omega) \qquad s.t. \quad \|(\mathbb{E}_\omega[f])(x) - (\mathbb{E}_\omega[f])(gx)\|_{L^2(\mathcal{X})}^2$$
$$\leq \varepsilon \, \mathbb{E}_g \left[ \|f(x) - f(gx)\|_{L^2(\mathcal{X})}^2 \right], \quad \forall f \in \mathcal{F}, g \in G.$$

**Example 9.** Consider the set of constant functions on the domain. This function class clearly satisfies all notions of symmetry for any group action, and thus $\mathsf{AC}^{\mathrm{ex}}(\mathcal{F}) = \mathsf{AC}^{\mathrm{wk}}(\mathcal{F}, \varepsilon) = \mathsf{AC}^{\mathrm{st}}(\mathcal{F}, \varepsilon) = 1$, for all $\epsilon > 0$, as the learner needs just one (trivial) query to achieve any of these symmetries.

### 3.4 Properties of Averaging Complexity

Before presenting the main result of the paper, we first review basic properties of averaging complexity in the following proposition.

**Proposition 10** (Properties of Averaging Complexity). *The following properties hold for the different notions of averaging complexity:*

- *The functions $\mathsf{AC}^{\mathrm{wk}}(\mathcal{F}, \varepsilon)$ and $\mathsf{AC}^{\mathrm{st}}(\mathcal{F}, \varepsilon)$ are non-increasing in $\varepsilon$.*

- *For all $\varepsilon > 0$, $\mathsf{AC}^{\mathrm{wk}}(\mathcal{F}, \varepsilon) \leq \mathsf{AC}^{\mathrm{st}}(\mathcal{F}, \varepsilon) \leq \mathsf{AC}^{\mathrm{ex}}(\mathcal{F}) \leq |G|$.*

- *For all $\varepsilon > 0$, $\mathsf{AC}^{\mathrm{wk}}(\mathcal{F}, 4\varepsilon) \leq \mathsf{AC}^{\mathrm{st}}(\mathcal{F}, 4\varepsilon) \leq \mathsf{AC}^{\mathrm{wk}}(\mathcal{F}, \varepsilon)$.*

- *If $\mathcal{F}_1 \subseteq \mathcal{F}_2$, then $\mathsf{AC}^{\mathrm{ex}}(\mathcal{F}_1) \leq \mathsf{AC}^{\mathrm{ex}}(\mathcal{F}_2)$. The same holds for $\mathsf{AC}^{\mathrm{wk}}$ and $\mathsf{AC}^{\mathrm{st}}$.*

The proof of Proposition 10 is deferred to Appendix B. The first two properties follow directly from the definition of averaging complexity and are obtained via the trivial averaging scheme (i.e., querying all $g \in G$). Intuitively, the last inequality illustrates that enforcing exact symmetry becomes more difficult as the class grows.

The proof of the third property is more challenging: it relates the strong and weak notions of approximate symmetry when the precision is relaxed by a constant factor. This observation allows us to focus, for simplicity, only on the notion of weak approximate symmetry.

## 4 Main Results

The main purpose of this paper is to study how various notions of averaging complexity relate to properties of the group action and the function class, and whether there is a fundamental separation between exact and approximate symmetry. Such a separation would show that approximate symmetry is, in an abstract setting, fundamentally easier to achieve.

### 4.1 Assumptions and Definitions

We note that any form of averaging complexity can always be upper bounded *linearly* by the group size via the trivial averaging scheme that queries all group elements $g \in G$. This motivates the question of when *sublinear* averaging complexity is achievable.

To this end, the role of the function class is crucial: trivial classes, such as the set of constant functions, always have trivial averaging complexity. To avoid pathological cases, we assume the following condition for the domain, group action, and function class:

**Assumption 11** (Faithful Group Action). For every non-identity element $g \in G$, there exist $f \in \mathcal{F}$ and $x \in \mathcal{X}$ such that $f(gx) \neq f(x)$.

This assumption excludes degenerate cases while remaining sufficiently general. We next define (symmetric) tensor powers of a function class, which we use later in our results.

**Definition 12** (Symmetric Tensor Powers of Function Spaces). *Let $\mathcal{F}$ be a finite-dimensional vector space of functions on a domain $\mathcal{X}$ and let $k \in \mathbb{N}$. Define*

$$\mathrm{Sym}^{\otimes k}(\mathcal{F}) := \mathrm{span}\Big\{ \prod_{i=1}^{k} f_i(x) : f_i \in \mathcal{F} \text{ for } i \in [k] \Big\}, \qquad \widetilde{\mathrm{Sym}}^{\otimes k}(\mathcal{F}) := \bigoplus_{\ell=0}^{k} \mathrm{Sym}^{\otimes \ell}(\mathcal{F}),$$

*where $\mathrm{Sym}^{\otimes 0}(\mathcal{F})$ is the one-dimensional space of constant functions on $\mathcal{X}$.*

The construction above uses the base function class $\mathcal{F}$ to form the enlarged class $\widetilde{\mathrm{Sym}}^{\otimes k}(\mathcal{F})$, which consists of linear combinations of pointwise products of up to $k$ functions from $\mathcal{F}$. In particular, $\mathrm{Sym}^{\otimes 1}(\mathcal{F}) = \mathcal{F}$, while higher orders $k \in \mathbb{N}$ correspond to degree-$k$ homogeneous polynomial features constructed from elements of $\mathcal{F}$.

A canonical example is $\mathcal{X} = \mathbb{R}^d$ with $\mathcal{F}$ the set of linear functions on $\mathbb{R}^d$. In this case, $\widetilde{\mathrm{Sym}}^{\otimes k}(\mathcal{F})$ is exactly the space of polynomials in $x$ of total degree at most $k$. Another example arises in kernel methods: starting from a base kernel (and its feature map), one may form polynomial feature expansions, which correspond to tensor powers of the base feature space and yield increased expressivity.

In this paper, symmetric tensor powers serve as a tool for proving lower bounds on the averaging complexity of enforcing exact symmetry. Our goal is to show that there exists a relatively small degree $k$ (i.e., low-order polynomial features) for which the required averaging complexity scales linearly in $|G|$.

## 4.2 An Exponential Separation

The main result of this paper is summarized in the following series of theorems.

**Theorem 13** (Averaging Complexity of Exact Symmetry Enforcement). *Under the above assumptions, for any function class $\mathcal{F}$ there exists an integer $\mathsf{K}$, for which we provide an explicit closed-form expression, such that the averaging complexity of exact symmetry enforcement is*

$$\mathsf{AC}^{\mathrm{ex}}\left(\widetilde{\mathrm{Sym}}^{\otimes k}(\mathcal{F})\right) = |G|, \quad \forall k \geq \mathsf{K}. \tag{4.1}$$

The proof of Theorem 13 is given in Appendix C. By definition of tensor powers, one has $\widetilde{\mathrm{Sym}}^{\otimes k}(\mathcal{F}) \subseteq \widetilde{\mathrm{Sym}}^{\otimes k'}(\mathcal{F})$ for any $k' \geq k$. Since averaging complexity is monotone with respect to inclusion of function classes (Proposition 10), the quantity $\mathsf{AC}^{\mathrm{ex}}\left(\widetilde{\mathrm{Sym}}^{\otimes k}(\mathcal{F})\right)$ is nondecreasing in $k \in \mathbb{N}$; intuitively, enforcing exact symmetry becomes harder as the class grows. At the same time, $\mathsf{AC}^{\mathrm{ex}}(\widetilde{\mathrm{Sym}}^{\otimes k}(\mathcal{F})) \leq |G|$ for all $k \in \mathbb{N}$. Therefore, to prove Theorem 13, it suffices to show that $\mathsf{AC}^{\mathrm{ex}}(\widetilde{\mathrm{Sym}}^{\otimes k}(\mathcal{F})) = |G|$ for $k = \mathsf{K}$.

Theorem 13 asserts that exact symmetry requires *linear* averaging complexity once polynomial features of degree $k = \mathsf{K}$ are included. A natural question is how to bound $\mathsf{K}$. To answer this question, we establish an explicit upper bound on $\mathsf{K}$, building on recent advances in algebra. In particular, letting $\rho$ denote the representation of $G$ on $\mathcal{F}$, we show that $\mathsf{K}$ admits the upper bound

$$\mathsf{K} \leq \min\left\{|G|, \sum_{\lambda \in \Lambda} M_\lambda - 1\right\}, \tag{4.2}$$

where

$$\Lambda := \bigcup_{g \in G}\left\{\text{eigenvalues of } \rho(g)\right\}, \qquad M_\lambda := \max_{g \in G}\left\{\text{multiplicity of } \lambda \text{ as an eigenvalue of } \rho(g)\right\}.$$

**Example 14.** Let $\mathcal{X} = \mathbb{R}^d$ and let $G = S_d$ act by permuting the coordinates of $x \in \mathbb{R}^d$. Let $\mathcal{F}$ be the class of all linear functions on $\mathbb{R}^d$. In this setting, for each $g \in G$, the matrix $\rho(g) \in \mathbb{R}^{d \times d}$ is the permutation matrix associated with $g$. If $g$ has cycle decomposition in $S_d$ with cycle lengths $(\ell_1, \ell_2, \ldots, \ell_t)$ satisfying $\sum_{j=1}^{t} \ell_j = d$, then the eigenvalues of $\rho(g)$ are

$$\exp\left(\frac{2\pi i p}{\ell_j}\right), \quad p = 0, 1, \ldots, \ell_j - 1, \quad j = 1, \ldots, t.$$

Moreover, if $\lambda$ is an eigenvalue of some $\rho(g)$, $g \in G$, with order $q$ (i.e., minimum $q \in \mathbb{N}$ such that $\lambda^q = 1$), then we have $M_\lambda = \lfloor \frac{d}{q} \rfloor$. A simple counting argument shows that

$$\sum_{\lambda \in \Lambda} M_\lambda = \sum_{q=1}^{d} \varphi(q) \left\lfloor \frac{d}{q} \right\rfloor = \frac{d(d+1)}{2},$$

where $\varphi(q) \in \mathbb{N}$ denotes Euler's totient function. Consequently, polynomial features of degree $\mathsf{K} = \frac{d(d+1)}{2} - 1$ already suffice to achieve linear averaging complexity for enforcing exact symmetry.

Next, we derive upper bounds on the averaging complexity of approximate symmetry, to compare with the exact case, which we already proved requires linear averaging complexity.

**Theorem 15** (Averaging Complexity of Approximate Symmetry Enforcement). *For any function class $\mathcal{F}$ and any $\varepsilon > 0$, the averaging complexities of weak and strong approximate symmetry enforcement satisfy*

$$\mathsf{AC}^{\mathrm{st}}(\mathcal{F}, \varepsilon) = \mathcal{O}\left(\frac{\log |G|}{\varepsilon}\right), \qquad \mathsf{AC}^{\mathrm{wk}}(\mathcal{F}, \varepsilon) = \mathcal{O}\left(\frac{\log |G|}{\varepsilon}\right),$$

*where the big-$\mathcal{O}$ notation hides universal constants.*

*Note*: The hidden constant in the big-$\mathcal{O}$ notation is at most $\frac{8}{3} \approx 2.67$ or $\frac{32}{3} \approx 10.67$ for weak or strong symmetry enforcement, respectively. The proof of Theorem 15 is given in Appendix D.

These bounds hold uniformly for all function classes and do not rely on Assumption 11 or on the use of tensor powers; they apply even beyond the tensor-power setting. Thus, the upper bounds for approximate symmetry enforcement are *universal*. In particular, they apply to the classes considered in Theorem 13, for which exact symmetry requires linear averaging complexity. Therefore, approximate symmetry enforcement needs only *logarithmic* averaging complexity (in $|G|$), yielding an *exponential* separation between the approximate and exact regimes.

> The averaging complexity of approximate symmetry enforcement (in the weak or strong sense) is $\mathcal{O}_\varepsilon(\log |G|)$, whereas exact symmetry requires complexity $|G|$. This yields an *exponential* separation between the two regimes, showing approximate symmetry is much easier to achieve in the abstract model of averaging complexity.

*Remark* 16. In our proofs we also show that the bounds in Theorem 15 are tight (up to constants). In other words, there exist instances that require at least $\Omega_\varepsilon(\log |G|)$ action queries (AQs) to achieve approximate symmetry. Details are provided in Appendix E.

## 5 PROOF SKETCH

We sketch the proofs of our main results. For Theorem 13, we show that averaging over the entire group is necessary to guarantee exact invariance. For Theorem 15, we outline how approximate symmetry yields a universal logarithmic averaging complexity. For background on representation theory, see (Fulton & Harris, 2013).

### 5.1 PROOF SKETCH FOR THEOREM 13

We first note that, by complete reducibility, any group representation $\rho$ can be decomposed into a direct sum of (complex) *irreducible representations (irreps)* as follows:

$$\rho \cong \bigoplus_{i=1}^{|\widehat{G}|} m_i \pi_i, \quad \forall i : m_i \in \mathbb{Z}_{\geq 0}, \tag{5.1}$$

where $\widehat{G}$ denotes the set of distinct irreps (equivalently, one per conjugacy class of the group), and $\pi_i$, $i = 1, 2, \ldots, |\widehat{G}|$, enumerate these irreps. Applying this to the representation induced on the function class $\mathcal{F}$ yields nonnegative integer coefficients $m_i$ for all $i$.

What happens to this decomposition when we extend it to tensor powers $\widetilde{\mathrm{Sym}}^{\otimes \mathsf{K}}(\mathcal{F})$ for some $\mathsf{K} \geq 1$? Let $\mathrm{Sym}^{\otimes k}(\rho)$ denote the induced representation on $\mathrm{Sym}^{\otimes k}(\mathcal{F})$ for $k \in [\mathsf{K}]$. In this case, for each $k \in [\mathsf{K}]$,

$$\mathrm{Sym}^{\otimes k}(\rho) \cong \bigoplus_{i=1}^{|\widehat{G}|} m_i^{(k)} \pi_i, \qquad \forall i : m_i^{(k)} \in \mathbb{Z}_{\geq 0}. \tag{5.2}$$

What happens if we have an averaging scheme $\omega(g)$ and apply it on a space with representation $\mathrm{Sym}^{\otimes k}(\rho)$? To analyze this, view $\omega : G \to \mathbb{R}$ as a *group signal* (a function on the group), and consider its *Fourier transform* $\widehat{\omega}$ defined by

$$\widehat{\omega}(\pi) = \sum_{g \in G} \omega(g) \pi(g)^*, \quad \forall \pi \in \widehat{G}, \tag{5.3}$$

where $*$ denotes the conjugate transpose (adjoint) of a complex-valued matrix.

Using standard facts from representation theory, one concludes that averaging for $\mathrm{Sym}^{\otimes k}(\mathcal{F})$, $k \in \mathsf{K}$, yields exactly symmetric functions if and only if

$$\forall i : \quad \exists k \in [\mathsf{K}] : m_i^{(k)} \neq 0 \implies \left( \widehat{\omega}(\pi_i) = 0 \ \text{ or } \ \pi_i \text{ is trivial} \right). \tag{5.4}$$

Therefore, the function $\widehat{\omega} : \widehat{G} \to \mathbb{C}$ must have *sparse* support whenever many irreps appear in the direct-sum decomposition of $\mathrm{Sym}^{\otimes k}(\rho)$, $k \in \mathsf{K}$. We claim that for $\mathsf{K}$ given in Equation 4.2, every nontrivial irrep appears in some $\mathrm{Sym}^{\otimes k}(\rho)$ with $k \in \mathsf{K}$. If this claim holds, then

$$\widehat{\omega}(\pi_i) = 0 \quad \text{for all } i \text{ with } \pi_i \text{ nontrivial}.$$

But this means the Fourier transform of $\omega$ vanishes everywhere except at the point corresponding to the trivial irrep. By Fourier inversion, $\omega$ must be the uniform measure on $G$; since it sums to one, $\omega(g) = \frac{1}{|G|}$ for all $g \in G$. Thus, any averaging scheme achieving exact symmetry requires access to $|G|$ action queries, as claimed.

## 5.2 Proof Sketch for Theorem 15

We adopt the same Fourier-analytic viewpoint on $\omega$ as in the previous subsection. To establish that averaging complexity $\mathcal{O}\left(\frac{\log |G|}{\varepsilon}\right)$ is achievable under approximate symmetry, it suffices to construct $\omega : G \to \mathbb{R}$ such that

$$\mathrm{size}(\omega) = \mathcal{O}\left(\frac{\log |G|}{\varepsilon}\right), \qquad \forall \pi \text{ nontrivial} : \ \|\widehat{\omega}(\pi)\|_{\mathrm{op}} \leq \varepsilon. \tag{5.5}$$

We use a probabilistic construction. Sample $n$ group elements independently and uniformly at random, and let $\Omega$ be their empirical distribution (we use a capital letter to emphasize that it is chosen at random). Form the block-diagonal matrix $\Xi := \bigoplus_{\pi \text{ nontrivial}} \widehat{\Omega}(\pi)$. Then $\mathbb{E}[\Xi] = 0$ and, for every nontrivial $\pi$, $\|\widehat{\Omega}(\pi)\|_{\mathrm{op}} \leq \|\Xi\|_{\mathrm{op}}$. Thus, it is enough to control the operator norm of a zero-mean random matrix. Standard large deviation bounds imply that, with high probability, $\|\Xi\|_{\mathrm{op}} \leq \varepsilon$ provided $n \geq c \frac{\log \dim(\Xi)}{\varepsilon}$ for a universal constant $c$. From representation theory, $\dim(\Xi) \leq |G|$, which yields the claimed $\mathcal{O}\left(\frac{\log |G|}{\varepsilon}\right)$ bound.

## 6 Conclusion and Future Directions

We presented a theoretical study of learning with symmetries, focusing on why *approximate* symmetry is both more convenient in practice and more reasonable for natural data. We introduced an abstract framework that defines the *averaging complexity* of enforcing exact or approximate symmetry as the minimum number of interactions with an oracle via action queries (AQs). Our main result shows an exponential separation: enforcing symmetry exactly can require linear complexity in $|G|$, whereas relaxing to approximate symmetry reduces the complexity to logarithmic in $|G|$, providing theoretical evidence for a sharp gap between the two regimes.

Several directions remain open. First, while this work focuses on finite groups, extending the framework and bounds to *infinite groups* is both natural and challenging, likely requiring ideas beyond those used here. Second, it would be valuable to leverage our abstract formulation, together with representation-theoretic methods, to analyze other theoretical problems in machine learning under symmetry, such as data augmentation. We leave these questions to future work.

ACKNOWLEDGMENTS

BT and MW were supported by NSF awards CBET-2112085 and DMS-2406905. MW also acknowledges support from an Alfred P. Sloan Fellowship in Mathematics and an Aramont Fellowship for Emerging Science Research.

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

# A BACKGROUND FOR PROOFS

This appendix collects the background used in our proofs. We briefly review finite groups, group actions, group representations, character theory, and Fourier analysis on finite groups (Serre, 1977; Isaacs, 1994; Fulton & Harris, 2013).

## A.1 GROUP THEORY

A *finite group* is a finite set $G$ equipped with a binary operation $\cdot : G \times G \to G$ satisfying:

- (Associativity) For all $g, h, k \in G$, $(g \cdot h) \cdot k = g \cdot (h \cdot k)$.
- (Identity) There exists an element $e \in G$ such that $e \cdot g = g \cdot e = g$ for all $g \in G$.
- (Inverses) For each $g \in G$, there exists $g^{-1} \in G$ with $g \cdot g^{-1} = g^{-1} \cdot g = e$.

Given a finite group $G$, we denote its identity element by $e$. For brevity, we omit the operation symbol and write $gh$ for $g \cdot h$.

The *order* of $G$ is the number of its elements, denoted $|G|$. For every integer $n \geq 1$, there exists a group of order $n$: the cyclic group $\mathbb{Z}/n\mathbb{Z} := \{0, 1, \ldots, n-1\}$ under addition modulo $n$.

A canonical example of a finite group is the *symmetric group* $S_d$, the group of all permutations of $d$ elements:
$$S_d := \big\{ \sigma : [d] \to [d] \,\big|\, \sigma \text{ is bijective} \big\},$$
with composition as the group operation. Here, we use the notation $[d] := \{1, 2, \ldots, d\}$ for $d \in \mathbb{N}$.

Define a relation $\sim$ on $G$ by $g \sim h \iff \exists s \in G : h = sgs^{-1}$. This is an equivalence relation on $G$. The *conjugacy class* of $g$ is $[g] := \{sgs^{-1} : s \in G\}$. The conjugacy classes $\{[g] : g \in G\}$ form a partition of $G$. Let $r$ denote the number of conjugacy classes of $G$. If $[g_1], \ldots, [g_r]$ are the distinct conjugacy classes, then
$$G = \bigsqcup_{i=1}^{r} [g_i].$$

Trivially, $r \leq |G|$. For commutative groups (i.e., $gh = hg$ for all $g, h \in G$), this bound is tight: $r = |G|$, since every conjugacy class is a singleton.

In contrast, for many noncommutative groups one has $r \ll |G|$. A canonical example is the symmetric group $S_d$, where conjugacy classes correspond to cycle type; hence $r = p(d)$, the partition number, which is far smaller than $|S_d| = d!$. Asymptotically, $\log p(d) = \Theta(\sqrt{d})$ while $\log |S_d| = \log(d!) = \Theta(d \log d)$. Thus, in this case we have $r \ll |S_d|$. Another canonical example is the dihedral group $D_{2n}$, the symmetries of a regular $n$-gon, which has $2n$ elements (rotations and reflections). It has $\frac{n+3}{2}$ conjugacy classes when $n$ is odd and $\frac{n}{2} + 3$ when $n$ is even; in particular, $r < |D_{2n}| = 2n$.

## A.2 GROUP ACTIONS AND FUNCTION SPACES

Let $\mathcal{X}$ be a topological space and $G$ a finite group. A (left) *group action* of $G$ on $\mathcal{X}$ is a map $\theta : G \times \mathcal{X} \to \mathcal{X}$ such that $\theta(gh, x) = \theta\big(g, \theta(h, x)\big)$ for all $g, h \in G$ and $x \in \mathcal{X}$, and the identity element of $G$ acts trivially (via the identity map $x \mapsto x$) on $\mathcal{X}$. For notational convenience, we write $gx := \theta(g, x)$. We consider only continuous actions: for each $g \in G$, the map $x \mapsto gx$ is a homeomorphism of $\mathcal{X}$ onto itself.

Let $\mathcal{X}$ be a topological space and let $\mathcal{B}(\mathcal{X})$ denote its Borel $\sigma$-algebra, making $(\mathcal{X}, \mathcal{B}(\mathcal{X}))$ a measurable space. Fix a reference measure $\mu$ on $\mathcal{X}$; all function spaces below are defined with respect to $\mu$. Without loss of generality, we assume the action of $G$ preserves the reference measure $\mu$, i.e., $\mu(gA) = \mu(A)$ for all measurable $A \subseteq \mathcal{X}$ and $g \in G$ (equivalently, $d\mu(gx) = d\mu(x)$ for all $g \in G$). For finite groups, this can always be arranged by averaging any reference measure $\mu_{\text{ref}}$ over $G$:
$$\mu(A) := \frac{1}{|G|} \sum_{g \in G} \mu_{\text{ref}} \big(g^{-1} A\big),$$

which is $G$-invariant. In many settings, there is also a canonical "uniform" choice (e.g., counting measure on finite sets or Haar/surface/Lebesgue measure on standard spaces) under which the usual actions are measure-preserving.

The space of square-integrable functions is

$$L^2(\mathcal{X}) := \left\{ f : \mathcal{X} \to \mathbb{C} \text{ measurable } : \|f\|_{L^2(\mathcal{X})}^2 := \int_{\mathcal{X}} |f(x)|^2 \, \mathrm{d}\mu(x) < \infty \right\}.$$

Let $\mathcal{F} \subseteq L^2(\mathcal{X})$ be a finite-dimensional subspace of continuous real-valued functions that is stable under $G$, i.e., $f(gx) \in \mathcal{F}$ for all $f \in \mathcal{F}$ and $g \in G$. The action of $G$ on $\mathcal{X}$ induces a (left) action on $\mathcal{F}$ by:

$$(gf)(x) := f(g^{-1}x) \in \mathcal{F}, \qquad \forall g \in G, \ f \in \mathcal{F}, \ x \in \mathcal{X}.$$

Recall that an action of $G$ on $\mathcal{U}$ (either $\mathcal{X}$ or $\mathcal{F}$) is *faithful* if and only if

$$\forall u \in \mathcal{U}, \quad gu = u \ \Rightarrow \ g \text{ is the identity element of } G.$$

In this paper, we always assume that the function class $\mathcal{F}$ satisfies Assumption 11: the action of $G$ on $\mathcal{F}$ is faithful. That is, for every non-identity group element $g \in G$, there exists a function $f \in \mathcal{F}$ and a point $x \in \mathcal{X}$ such that $f(gx) \neq f(x)$. Note that Assumption 11 implies that the action of $G$ on $\mathcal{X}$ is also faithful: if a non-identity $g \in G$ fixed every $x \in \mathcal{X}$, then we would have $f(gx) = f(x)$ for all $f \in \mathcal{F}$, contradicting Assumption 11.

## A.3 GROUP REPRESENTATION THEORY

We use several notions from representation theory to establish our main results. This appendix reviews group representation theory in detail, with a particular focus on finite groups. For a comprehensive reference, see (Fulton & Harris, 2013).

Let $G$ be a finite group and let $V$ be a finite-dimensional (real or complex) inner-product space. Let $GL(V)$ denote the group of invertible linear maps $\psi : V \to V$ (under composition). A (linear) group representation is a group homomorphism $\rho : G \to GL(V)$, meaning $\rho(gh) = \rho(g)\rho(h)$ for all $g, h \in G$. After fixing a basis for $V$, each $\rho(g)$ can be viewed as a matrix in $\mathbb{R}^{\dim V \times \dim V}$ (or $\mathbb{C}^{\dim V \times \dim V}$). In other words, a representation "encodes" group elements by matrices so that group multiplication corresponds to matrix multiplication. For example, the *trivial* representation is defined as $\rho(g) = 1 \in \mathbb{R}$ for all $g \in G$.

In this paper, we assume representations are orthogonal (or unitary in the complex case): $\rho(g)^\top \rho(g) = I$ (respectively, $\rho(g)^* \rho(g) = I$) for all $g \in G$. Equivalently, $\langle \rho(g)u, \rho(g)v \rangle_V = \langle u, v \rangle_V$ for all $u, v \in V$. This assumption holds without loss of generality in our setting: when $V = \mathcal{F} \subseteq L^2(\mathcal{X})$ with the $L^2(\mathcal{X})$ inner product and the action is measure-preserving (i.e., $d\mu(gx) = d\mu(x)$), the induced action is orthogonal (unitary). Indeed, for any $f, f' \in \mathcal{F}$ and $g \in G$,

$$\langle \rho(g)f, \rho(g)f' \rangle_{L^2(\mathcal{X})} = \int_{\mathcal{X}} f(g^{-1}x) \, \overline{f'(g^{-1}x)} \, d\mu(x)$$

$$= \int_{\mathcal{X}} f(x) \, \overline{f'(x)} \, d\mu(gx)$$

$$= \int_{\mathcal{X}} f(x) \, \overline{f'(x)} \, d\mu(x) = \langle f, f' \rangle_{L^2(\mathcal{X})}.$$

Two representations $\rho$ and $\rho'$ of $G$ on $V$ are *equivalent* if there exists an orthogonal (unitary) matrix $U \in \mathbb{R}^{\dim V \times \dim V}$ (resp. $\mathbb{C}^{\dim V \times \dim V}$) such that $U\rho(g) = \rho'(g)U$ for all $g \in G$. A representation $\rho$ is *reducible* if it is equivalent to a nontrivial block-diagonal representation (simultaneously for all $g \in G$); otherwise, $\rho$ is *irreducible* (abbreviated "*irrep*," which we use throughout, consistent with standard representation-theory terminology).

Irreps are fundamental building blocks of representations. The main important result in representation theory of finite group is that any representation can be decomposed into irreps.

**Theorem 17** (Maschke's Theorem). *Let $G$ be a finite group. Over $\mathbb{R}$ or $\mathbb{C}$, every finite-dimensional representation of $G$ decomposes as a direct sum of irreducible representations.*

In particular, a finite group $G$ has only finitely many irreducible representations (up to equivalence), which we index as $\pi_i$ for $i \in [r]$, where $r$ is their number. Any representation $\rho$ of $G$ on a finite-dimensional space $V$ decomposes as

$$\rho \cong \bigoplus_{i \in [r]} m_i \pi_i, \qquad m_i \in \mathbb{Z}_{\geq 0}.$$

Here "$\oplus$" means that, after a change of basis (equivalence of representations), all matrices $\rho(g)$ become block diagonal simultaneously, with $m_i$ blocks each equivalent to $\pi_i$. The nonnegative integers $m_i$ are the *multiplicities* of the irreps $\pi_i$.

**Example 18.** Let $\rho$ be the natural permutation representation of the symmetric group $S_d$ on $\mathbb{R}^d$, acting by coordinate permutation:

$$\rho(\sigma)x = P_\sigma x, \qquad \sigma \in S_d,$$

where $P_\sigma$ is the permutation matrix of $\sigma$. This representation is reducible: the subspace $\mathrm{span}\{\mathbf{1}\}$ (with $\mathbf{1} = (1, \ldots, 1)$) is $S_d$-invariant (the *trivial* representation $\pi_1$), and its orthogonal complement $\{x \in \mathbb{R}^d : \sum_{i=1}^d x_i = 0\}$ is also $S_d$-invariant (the *standard* representation $\pi_2$) of dimension $d - 1$. In fact, $\rho \cong \pi_1 \oplus \pi_2$, and both are irreducible.

What do we know about irreps of a finite group $G$? If we index them by $\pi_i$, $i \in [r]$, then $r$ equals the number of conjugacy classes of $G$. We write

$$\widehat{G} := \{\pi : \pi \text{ is an irrep of } G\}, \qquad r = |\widehat{G}| = \text{the number of conjugacy classes of } G.$$

In particular, $|\widehat{G}| \leq |G|$; for commutative groups this is tight, $|\widehat{G}| = |G|$, while for noncommutative groups one has $|\widehat{G}| < |G|$, and in many cases even $|\widehat{G}| \ll |G|$ (e.g., for the symmetric group $S_d$, as we discussed before).

We now focus on complex irreducible representations of a finite group $G$. For an irrep $\pi$, let its dimension be $d_\pi \in \mathbb{N}$; thus $\pi(g) \in \mathbb{C}^{d_\pi \times d_\pi}$ for all $g \in G$. For commutative groups, all irreps are one-dimensional: $d_\pi = 1$ for every $\pi \in \widehat{G}$. In contrast, noncommutative groups admit higher-dimensional irreps.

For the complex irreps of a finite group, we have the identity:

$$|G| = \sum_{\pi \in \widehat{G}} d_\pi^2.$$

**Example 19.** For the symmetric group $S_d$, we have $|S_d| = d! = \exp\left(\Theta(d \log d)\right)$, while $|\widehat{S_d}| = p(d) = \exp\left(\Theta(\sqrt{d})\right)$, where $p(d)$ is the number of integer partitions of $d$. In this case,

$$d! = \sum_{\pi \in \widehat{S_d}} d_\pi^2 = \underbrace{1}_{\text{trivial irrep}} + \underbrace{(d-1)^2}_{\text{standard irrep}} + \text{ other terms.}$$

Thus, many irreps exist beyond those appearing in the natural permutation representation (trivial and standard), even though the natural permutation representation is faithful. In other words, faithfulness does not imply that a representation contains all irreps. In this case, several irreps have dimensions growing superpolynomially in $d$. A complete classification of $\widehat{S_d}$ is given by the partitions of $d$.

## A.4 EQUIVALENCE BETWEEN INVARIANCE AND EQUIVARIANCE

Adopting the previous definitions and notations, let $V$ denote a (complex-valued) finite-dimensional representation of $G$ and let us consider the space $\mathcal{F}(V) := \mathrm{span}\{vf : v \in V, f \in \mathcal{F}\} = \mathcal{F} \otimes V$. In other words, for any function $f : \mathcal{X} \to \mathbb{C}$ and any vector $v \in V$, one can define $vf : \mathcal{X} \to V$ in a natural way. Moreover, the group $G$ acts on $\mathcal{F}(V) = \mathcal{F} \otimes V$ naturally via the tensor product of the two diagonal representations.

Now consider *equivariant* functions within $\mathcal{F}(V)$, which we denote via $\mathcal{F}(V)^G$ (as functions from $\mathcal{X}$ to $V$). Such functions are defined as $\varphi \in \mathcal{F}(V)$ such that $\varphi(gx) = g\varphi(x)$ for all $x, g$. In other

words, we must have that $g\varphi\left(g^{-1}x\right) = \varphi(x)$ for all $x, g$. This means that $\varphi \in \mathcal{F}(V)$ is equivariant if and only if it is an invariant element of $\mathcal{F} \otimes V$.

In other words, if we consider the space of linear functions on $\widetilde{\mathcal{X}} := \mathcal{F} \otimes V$, then equivariant functions from $\mathcal{X}$ to $V$ are precisely invariant functions from $\widetilde{\mathcal{X}}$ to $\mathbb{C}$. This completes the proof of correspondence.

As a result, all the claims and proofs in the paper will apply to the equivariant function classes after applying appropriate changes. In particular, the exponential separation will again apply to such cases, with no further assumptions.

## A.5 FOURIER ANALYSIS ON FINITE GROUPS

The theory of *Fourier analysis on finite groups* is essential for the results in this paper. It is built on group representation theory and has numerous applications, including signal processing on groups.

**Definition 20** (Fourier Transform on Finite Groups). *Let $G$ be a finite group and let $\omega : G \to \mathbb{C}$ be a (complex-valued) signal on $G$. The* Fourier transform *of $\omega$ is the collection of matrices indexed by irreps $\pi \in \widehat{G}$,*

$$\widehat{\omega}(\pi) := \sum_{g \in G} \omega(g)\,\pi(g)^*, \qquad \pi \in \widehat{G}, \tag{A.1}$$

*where $^*$ denotes the conjugate transpose. This means that while the signal is supported on the group $G$, its Fourier transform is supported on $\widehat{G}$ (one matrix per irrep).*

Many natural properties of fourier transform on $\mathbb{C}^d$ also hold for finite groups. For instance, we have *Fourier inversion formula*:

$$\omega(g) = \frac{1}{|G|} \sum_{\pi \in \widehat{G}} d_\pi \operatorname{Tr}\left(\widehat{\omega}(\pi)\,\pi(g)\right). \tag{A.2}$$

Moreover, for any $\omega, \eta : G \to \mathbb{C}$,

$$\sum_{g \in G} \omega(g)\,\overline{\eta(g)} = \frac{1}{|G|} \sum_{\pi \in \widehat{G}} d_\pi \operatorname{Tr}\left(\widehat{\omega}(\pi)\,\widehat{\eta}(\pi)^*\right). \tag{A.3}$$

If we set $\eta = \omega$, we obtain the *Plancherel formula*:

$$\sum_{g \in G} |\omega(g)|^2 = \frac{1}{|G|} \sum_{\pi \in \widehat{G}} d_\pi \,\|\widehat{\omega}(\pi)\|_{\mathrm{F}}^2, \tag{A.4}$$

where $\|\cdot\|_{\mathrm{F}}$ denotes the Frobenius norm of matrices.

**Example 21.** Consider a group signal $\omega : G \to \mathbb{C}$ with the property

$$\widehat{\omega}(\pi) = 0, \quad \text{for all nontrivial } \pi \in \widehat{G}.$$

What does this *sparsity* of the Fourier transform imply? By the inversion formula,

$$\omega(g) = \frac{1}{|G|} \sum_{\pi \in \widehat{G}} d_\pi \operatorname{Tr}\left(\widehat{\omega}(\pi)\,\pi(g)\right) \tag{A.5}$$

$$= \frac{1}{|G|} \operatorname{Tr}\left(\widehat{\omega}(\pi_{\mathrm{triv}})\,\pi_{\mathrm{triv}}(g)\right) \tag{A.6}$$

$$= \frac{1}{|G|} \widehat{\omega}(\pi_{\mathrm{triv}}), \tag{A.7}$$

for all $g \in G$, where $\pi_{\mathrm{triv}}$ is the one-dimensional trivial irrep. Hence $\omega$ must be constant on $G$. If, in addition, $\|\omega\|_{\ell_1(G)} = \sum_{g \in G} \omega(g) = 1$, then necessarily

$$\omega(g) = \frac{1}{|G|} \quad \text{for all } g \in G, \tag{A.8}$$

i.e., $\omega$ is the uniform distribution on $G$. We will use this fact later to obtain our main result on the linearity of averaging complexity for exact symmetry enforcement.

### A.6 Invariant Subspaces and Fourier Analysis (Exact Symmetry)

In this subsection, we review core properties of group actions on function spaces and how they relate to the subspace of exactly symmetric functions. These tools are essential in our proofs.

Consider a finite-dimensional vector space $\mathcal{F}$ of complex-valued functions on the domain $\mathcal{X}$, as before. Not all functions in $\mathcal{F}$ are exactly symmetric; the *invariant subspace*

$$\mathcal{F}_G := \{f \in \mathcal{F} : gf = f, \ \forall g \in G\} \subseteq \mathcal{F} \tag{A.9}$$

is, in nontrivial cases, a proper subset of $\mathcal{F}$.

Let $\rho$ denote the representation of the finite group $G$ induced on $\mathcal{F}$. We write its decomposition as

$$\rho \cong \bigoplus_{i \in [r]} m_i \, \pi_i, \qquad m_i \in \mathbb{Z}_{\geq 0}. \tag{A.10}$$

How can one relate the invariant subspace $\mathcal{F}_G$ to the decomposition of $\rho$ into the irreps of $G$? To this end, consider the uniform signal $\omega(g) = \frac{1}{|G|}$ for all $g \in G$, and compute its Fourier transform:

$$\widehat{\omega}(\pi) = \sum_{g \in G} \omega(g) \, \pi(g)^* = \frac{1}{|G|} \sum_{g \in G} \pi(g)^* = \mathbb{E}_g[\pi(g)^*], \quad \forall \pi \in \widehat{G}. \tag{A.11}$$

However, using the Fourier inversion formula, we have shown in the previous section that for the uniform signal, $\widehat{\omega}(\pi) = 0$ for any nontrivial $\pi \in \widehat{G}$. Therefore, we conclude that

$$\mathbb{E}_g[\pi(g)^*] = \begin{cases} 0 \in \mathbb{R}^{d_\pi \times d_\pi}, & \text{if } \pi \text{ is nontrivial,} \\ 1 \in \mathbb{R}, & \text{if } \pi \text{ is trivial.} \end{cases} \tag{A.12}$$

Note that after a change of coordinates (i.e., choosing an appropriate basis of $\mathcal{F}$), we can write the group representation $\rho$ in block-diagonal form:

$$\rho(g) = \bigoplus_{i \in [r]} \big(I_{m_i} \otimes \pi_i(g)\big) \in \mathbb{R}^{\dim(\mathcal{F}) \times \dim(\mathcal{F})}, \qquad \forall g \in G, \tag{A.13}$$

where $I_{m_i} \in \mathbb{R}^{m_i \times m_i}$ denotes the identity matrix for each $i \in [r]$. Therefore,

$$\mathbb{E}_g\big[\rho(g)\big] = \bigoplus_{i \in [r]} \big(I_{m_i} \otimes \mathbb{E}_g\big[\pi_i(g)\big]\big) = I_{m_{\text{triv}}} \oplus 0 \oplus 0 \oplus \cdots, \tag{A.14}$$

where we have indexed the trivial irrep by $i = 1$. Note that, according to the above derivation, we also obtain

$$m_{\text{triv}} = \text{Tr}\big(\mathbb{E}_g\big[\rho(g)\big]\big) = \mathbb{E}_g\big[\text{Tr}\big(\rho(g)\big)\big], \tag{A.15}$$

where the quantities $\text{Tr}\big(\rho(g)\big)$, for $g \in G$, are commonly referred to as the *characters* of the group representation $\rho$.

Define $\Pi := \mathbb{E}_g[\rho(g)]$. For the basis of $\mathcal{F}$ above that block-diagonalizes $\rho$ (the "appropriate" basis), identify each $f \in \mathcal{F}$ with its coefficient vector $\boldsymbol{f} \in \mathbb{C}^{\dim(\mathcal{F})}$. Then,

$$\forall g \in G: \qquad gf \longleftrightarrow \rho(g)\,\boldsymbol{f}. \tag{A.16}$$

Then

$$f \in \mathcal{F}_G \iff \forall g \in G: \ gf = f \tag{A.17}$$

$$\iff \forall g \in G: \ \rho(g)\,\boldsymbol{f} = \boldsymbol{f} \tag{A.18}$$

$$\iff \frac{1}{|G|} \sum_{g \in G} \rho(g)\,\boldsymbol{f} = \boldsymbol{f} \tag{A.19}$$

$$\iff \Pi\,\boldsymbol{f} = \boldsymbol{f} \tag{A.20}$$

$$\iff \boldsymbol{f} = \big(\boldsymbol{f}_{\text{triv}}, \boldsymbol{0}, \boldsymbol{0}, \dots\big) \quad \text{(i.e., all nontrivial blocks are zero).} \tag{A.21}$$

In particular, $\Pi^2 = \Pi$ and $\Pi^* = \Pi$, so $\Pi$ is the orthogonal projector onto its image, which is $\mathcal{F}_G$, and thus

$$\dim(\mathcal{F}_G) \;=\; \mathrm{rank}\,(\Pi) \;=\; m_{\mathrm{triv}} = \mathbb{E}_g\left[\mathrm{Tr}\left(\rho(g)\right)\right]. \tag{A.22}$$

Note that we used the fact that

$$\forall\, g \in G: \; \rho(g)\,\boldsymbol{f} = \boldsymbol{f} \;\iff\; \frac{1}{|G|}\sum_{g\in G}\rho(g)\,\boldsymbol{f} = \boldsymbol{f}. \tag{A.23}$$

This is proved as follows. If $\rho(g)\boldsymbol{f} = \boldsymbol{f}$ for all $g \in G$, summing the equalities yields $\frac{1}{|G|}\sum_{g\in G}\rho(g)\,\boldsymbol{f} = \boldsymbol{f}$. Conversely, suppose $\frac{1}{|G|}\sum_{g\in G}\rho(g)\,\boldsymbol{f} = \boldsymbol{f}$. Then for any $g \in G$,

$$\rho(g)\,\boldsymbol{f} = \rho(g)\,\frac{1}{|G|}\sum_{g'\in G}\rho(g')\,\boldsymbol{f} = \frac{1}{|G|}\sum_{g'\in G}\rho(g)\rho(g')\,\boldsymbol{f} \tag{A.24}$$

$$= \frac{1}{|G|}\sum_{g'\in G}\rho(gg')\,\boldsymbol{f} = \frac{1}{|G|}\sum_{g''\in G}\rho(g'')\,\boldsymbol{f} = \boldsymbol{f}, \tag{A.25}$$

which completes the proof.

## A.7 Invariant Subspaces and Fourier Analysis (Approximate Symmetry)

We now relate weak approximate symmetry of a function $f \in \mathcal{F}$ to its coefficient vector $\boldsymbol{f} \in \mathbb{R}^m$ with $m := \dim(\mathcal{F})$. We have already shown that if $f$ is exactly symmetric, then its coefficient vector has the form $\boldsymbol{f} = \left(\boldsymbol{f}_{\mathrm{triv}}, \boldsymbol{0}, \boldsymbol{0}, \dots\right) \in \mathbb{R}^m$. In general, we have $\boldsymbol{f} = \left(\boldsymbol{f}_{\mathrm{triv}}, \boldsymbol{f}_{\mathrm{non}}\right) \in \mathbb{R}^m$ where $\mathcal{F} = \mathcal{F}_G \oplus \mathcal{F}_G^\perp$ and $m_{\mathrm{triv}} := \dim(\mathcal{F}_G)$ and $m_{\mathrm{non}} := \dim(\mathcal{F}_G^\perp)$.

For a weakly symmetric function $f \in \mathcal{F}$ with parameter $\epsilon > 0$, we have

$$\mathbb{E}_g\left[\int_{\mathcal{X}} \left|(\mathbb{E}_\omega[f])(x) - (\mathbb{E}_\omega[f])(gx)\right|^2 d\mu(x)\right] \;\leq\; \epsilon\,\mathbb{E}_g\left[\int_{\mathcal{X}} |f(x) - f(gx)|^2\, d\mu(x)\right]. \tag{A.26}$$

Note that, using measure preservation of the group action on $\mathcal{X}$ and the definition of $\Pi$,

$$\mathbb{E}_g\left[\int_{\mathcal{X}} |f(x) - f(gx)|^2\, d\mu(x)\right] = \mathbb{E}_g\Big[\int_{\mathcal{X}} |f(x)|^2\, d\mu(x) + \int_{\mathcal{X}} |f(gx)|^2\, d\mu(x) \tag{A.27}$$

$$- 2\int_{\mathcal{X}} f(x) f(gx)\, d\mu(x)\Big] \tag{A.28}$$

$$= 2\,\|\boldsymbol{f}\|_2^2 \;-\; 2\int_{\mathcal{X}} f(x)\,\mathbb{E}_g[f(gx)]\, d\mu(x) \tag{A.29}$$

$$= 2\,\|\boldsymbol{f}\|_2^2 \;-\; 2\,\langle \boldsymbol{f},\, \Pi\boldsymbol{f}\rangle \tag{A.30}$$

$$= 2\,\|\boldsymbol{f}\|_2^2 \;-\; 2\|\boldsymbol{f}_{\mathrm{triv}}\|_2^2, \tag{A.31}$$

$$= 2\,\|\boldsymbol{f}_{\mathrm{non}}\|_2^2. \tag{A.32}$$

Therefore, we conclude that

$$\mathbb{E}_\omega[.] \text{ is } \epsilon\text{-weakly approx. symm.} \iff \left\|(\mathbb{E}_\omega \boldsymbol{f})_{\mathrm{non}}\right\|_2^2 \leq \epsilon\,\|\boldsymbol{f}_{\mathrm{non}}\|_2^2, \quad \forall f \in \mathcal{F} \tag{A.33}$$

## A.8 A Note on the Relationship with Sample Complexity under Symmetries

In this subsection, we briefly review how the results derived in this paper relate to the sample complexity of learning under symmetries. Let $\mathcal{F} \subseteq L^2(\mathcal{X})$ be a finite-dimensional vector space of continuous functions on $\mathcal{X}$. Draw samples $x_i \in \mathcal{X}$, $i \in [n]$, i.i.d. from a reference probability measure $\mu$ on $\mathcal{X}$. Let $f^\star \in \mathcal{F}$ be a target function and observe labels

$$\forall\, i \in [n]: \quad y_i = f^\star(x_i) + \epsilon_i, \tag{A.34}$$

where the noise terms $\epsilon_i$ are independent and identically distributed with law $\mathcal{N}(0, \sigma^2)$.

The empirical risk minimizer (ERM) is

$$\widehat{f}_{\mathrm{ERM}} \;:=\; \arg\min_{f \in \mathcal{F}} \Big\{ \frac{1}{2n} \sum_{i=1}^{n} (f(x_i) - y_i)^2 \Big\}. \tag{A.35}$$

The *excess population risk* (or generalization error) of an estimator $\widehat{f}$ is defined as

$$\mathcal{R}(\widehat{f}) \;:=\; \mathbb{E}\Big[ \|\widehat{f} - f^\star\|_{L^2(\mathcal{X})}^2 \Big], \tag{A.36}$$

where the expectation is taken over the noise terms. Since $\widehat{f}$ is constructed from the training samples, the quantity $\mathcal{R}(\widehat{f})$ is itself a random variable. In our analysis, we derive high-probability bounds on $\mathcal{R}(\widehat{f})$ with respect to the randomness of the sampled inputs.

When learning under (exact) symmetries, we assume that $f^\star$ is symmetric: $g f^\star = f^\star$ for all $g \in G$. It is then desirable to encode the known symmetry of $f^\star$ in the ERM output via exact or approximate symmetrization. Motivated by this, define the *exactly symmetrized* and *weakly symmetrized* ERM estimators by

$$\widehat{f}_{\mathrm{ERM}}^{\mathrm{ex}}(x) \;:=\; \frac{1}{|G|} \sum_{g \in G} \widehat{f}_{\mathrm{ERM}}(g^{-1}x), \tag{A.37}$$

$$\widehat{f}_{\mathrm{ERM}}^{\mathrm{wk}}(x) \;:=\; \big(\mathbb{E}_\omega[\widehat{f}_{\mathrm{ERM}}]\big)(x) \;=\; \sum_{g \in G} \omega(g)\, \widehat{f}_{\mathrm{ERM}}(g^{-1}x), \tag{A.38}$$

where $\omega : G \to \mathbb{R}$ is an averaging scheme chosen to ensure $\epsilon$-weak approximate symmetry.

Let $\varphi_j(x)$, for $j = 1, 2, \ldots, \dim(\mathcal{F})$, be an $L^2(\mathcal{X})$-orthonormal basis for $\mathcal{F}$, and let $\Phi(x) := (\varphi_1(x), \ldots, \varphi_{\dim(\mathcal{F})}(x))^\top$ denote the corresponding feature vector. For any $f \in \mathcal{F}$ with coefficient vector $\boldsymbol{f} \in \mathbb{R}^{\dim(\mathcal{F})}$, we have $f(x) = \langle \boldsymbol{f}, \Phi(x) \rangle$.

Given samples $x_1, \ldots, x_n$, let $X \in \mathbb{R}^{n \times \dim(\mathcal{F})}$ be the design matrix with $X_{ij} = \varphi_j(x_i)$ for each $\ell, i$. Let $\boldsymbol{y} = (y_i)_{i=1}^{n} = X\boldsymbol{f}^\star + \boldsymbol{\epsilon} \in \mathbb{R}^n$. Then, the ERM problem can be written as

$$\widehat{\boldsymbol{f}}_{\mathrm{ERM}} \;:=\; \arg\min_{\boldsymbol{f} \in \mathbb{R}^{\dim(\mathcal{F})}} \frac{1}{2n} \|X\boldsymbol{f} - \boldsymbol{y}\|_2^2 \quad\Longrightarrow\quad \widehat{\boldsymbol{f}}_{\mathrm{ERM}} = (X^\top X)^{-1} X^\top \boldsymbol{y}, \tag{A.39}$$

assuming $X$ has full column rank.

The excess population risk of ERM (with no symmetry enforcement) can be written as

$$\mathcal{R}(\widehat{f}_{\mathrm{ERM}}) := \mathbb{E}\big[\|\widehat{f}_{\mathrm{ERM}} - f^\star\|_{L^2(\mathcal{X})}^2\big] \;=\; \mathbb{E}\big[\|\widehat{\boldsymbol{f}}_{\mathrm{ERM}} - \boldsymbol{f}^\star\|_2^2\big] \tag{A.40}$$

$$= \mathbb{E}\big[\|(X^\top X)^{-1} X^\top (X\boldsymbol{f}^\star + \boldsymbol{\epsilon}) - \boldsymbol{f}^\star\|_2^2\big] \tag{A.41}$$

$$= \mathbb{E}\big[\|(X^\top X)^{-1} X^\top \boldsymbol{\epsilon}\|_2^2\big] \tag{A.42}$$

$$= \mathbb{E}\big[\boldsymbol{\epsilon}^\top X (X^\top X)^{-2} X^\top \boldsymbol{\epsilon}\big] \tag{A.43}$$

$$= \sigma^2 \operatorname{Tr}\big(X(X^\top X)^{-2} X^\top\big) \tag{A.44}$$

$$= \sigma^2 \operatorname{Tr}\big((X^\top X)^{-1}\big) \tag{A.45}$$

$$= \frac{\sigma^2}{n} \operatorname{Tr}\bigg( \Big(\frac{1}{n} X^\top X\Big)^{-1} \bigg) \tag{A.46}$$

$$= \frac{\sigma^2}{n} \operatorname{Tr}\bigg( \Big(\frac{1}{n} \sum_{i=1}^{n} \Phi(x_i)\Phi(x_i)^\top\Big)^{-1} \bigg), \tag{A.47}$$

where we used the cyclic property of the trace and the fact that $\boldsymbol{\epsilon} \sim \mathcal{N}(0, \sigma^2 I_n)$. Now, let us study the excess population risk of exact symmetry enforcement via group averaging. Let $\Pi$ denote the projection operator, as before. Note that $\Pi \boldsymbol{f}^\star = \boldsymbol{f}^\star$ and $\Pi^* = \Pi$. Moreover, $\operatorname{rank}(\Pi) = m_{\mathrm{triv}}$.

Then

$$\mathcal{R}(\widehat{f}_{\mathrm{ERM}}^{\mathrm{ex}}) \ := \ \mathbb{E}\big[\|\widehat{f}_{\mathrm{ERM}}^{\mathrm{ex}} - f^\star\|_{L^2(\mathcal{X})}^2\big] \ = \ \mathbb{E}\big[\|\Pi\widehat{\boldsymbol{f}}_{\mathrm{ERM}} - \boldsymbol{f}^\star\|_2^2\big] \tag{A.48}$$

$$= \mathbb{E}\big[\|\Pi(X^\top X)^{-1}X^\top(X\boldsymbol{f}^\star + \boldsymbol{\epsilon}) - \boldsymbol{f}^\star\|_2^2\big] \tag{A.49}$$

$$= \mathbb{E}\big[\|\Pi(X^\top X)^{-1}X^\top\boldsymbol{\epsilon}\|_2^2\big] \tag{A.50}$$

$$= \mathbb{E}\big[\boldsymbol{\epsilon}^\top X(X^\top X)^{-1}\Pi(X^\top X)^{-1}X^\top\boldsymbol{\epsilon}\big] \tag{A.51}$$

$$= \sigma^2 \operatorname{Tr}\big(X(X^\top X)^{-1}\Pi(X^\top X)^{-1}X^\top\big) \tag{A.52}$$

$$= \sigma^2 c \operatorname{Tr}\big(\Pi(X^\top X)^{-1}\big) \tag{A.53}$$

$$= \frac{\sigma^2}{n} \operatorname{Tr}\left(\Pi\left(\frac{1}{n}X^\top X\right)^{-1}\right) \tag{A.54}$$

$$= \frac{\sigma^2}{n} \operatorname{Tr}\left(\Pi\left(\frac{1}{n}\sum_{i=1}^n \Phi(x_i)\Phi(x_i)^\top\right)^{-1}\right). \tag{A.55}$$

Using the matrix Chernoff inequality (Tropp, 2012; Vershynin, 2018), and assuming $\sup_{x\in\mathcal{X}} \|\Phi(x)\|_2^2 \le c_0$, we obtain that with probability at least $1-\delta$,

$$\frac{1}{2}\,I_m \ \preceq \ \left(\frac{1}{n}\sum_{i=1}^n \Phi(x_i)\Phi(x_i)^\top\right)^{-1} \ \preceq \ \frac{3}{2}\,I_m, \qquad \forall\, n \ge c_0\,c_1 \log\left(\frac{m}{\delta}\right), \tag{A.56}$$

for an absolute constant $c_1$. Note that $c_0 \ge m$. Consequently, with the same probability,

$$\mathcal{R}(\widehat{f}_{\mathrm{ERM}}) \ = \ \Theta\left(\frac{\sigma^2 m}{n}\right), \qquad \mathcal{R}(\widehat{f}_{\mathrm{ERM}}^{\mathrm{ex}}) \ = \ \Theta\left(\frac{\sigma^2 m_{\mathrm{triv}}}{n}\right), \tag{A.57}$$

where $m = \dim(\mathcal{F})$ and $m_{\mathrm{triv}} = \dim(\mathcal{F}_G)$.

Finally, to study $\mathcal{R}\big(\widehat{f}_{\mathrm{ERM}}^{\mathrm{wk}}\big)$, note that a given averaging scheme $\omega : G \to \mathbb{R}$ induces a linear operator $\mathbb{E}_\omega : \mathcal{F} \to \mathcal{F}$; with a slight abuse of notation, we use the same symbol for its action on coefficient vectors.

Note that

$$\mathcal{R}\left(\widehat{f}_{\mathrm{ERM}}^{\mathrm{wk}}\right) \ := \ \mathbb{E}\big[\|\widehat{f}_{\mathrm{ERM}}^{\mathrm{wk}} - f^\star\|_{L^2(\mathcal{X})}^2\big] \ = \ \mathbb{E}\big[\|\widehat{\boldsymbol{f}}_{\mathrm{ERM}}^{\mathrm{wk}} - \boldsymbol{f}^\star\|_2^2\big] \tag{A.58}$$

$$= \mathbb{E}\big[\|\mathbb{E}_\omega\widehat{\boldsymbol{f}}_{\mathrm{ERM}} - \boldsymbol{f}^\star\|_2^2\big] \tag{A.59}$$

$$\le 2\,\mathbb{E}\big[\|\mathbb{E}_\omega\widehat{\boldsymbol{f}}_{\mathrm{ERM}} - \Pi\widehat{\boldsymbol{f}}_{\mathrm{ERM}}\|_2^2\big] + 2\,\mathbb{E}\big[\|\Pi\widehat{\boldsymbol{f}}_{\mathrm{ERM}} - \boldsymbol{f}^\star\|_2^2\big] \tag{A.60}$$

$$= 2\,\mathbb{E}\big[\|\mathbb{E}_\omega\widehat{\boldsymbol{f}}_{\mathrm{ERM}} - \Pi\widehat{\boldsymbol{f}}_{\mathrm{ERM}}\|_2^2\big] + \Theta\left(\frac{\sigma^2 m_{\mathrm{triv}}}{n}\right), \tag{A.61}$$

where we used the previous derivation of the high-probability excess population risk under exact symmetry enforcement. To upper bound the first term, note that the invariant subspace $\mathcal{F}_G$ is fixed by the linear operator $\mathbb{E}_\omega$:

$$\forall\, f \in \mathcal{F}_G \quad \Longrightarrow \quad \mathbb{E}_\omega[f] = f, \tag{A.62}$$

since $gf = f$ for all $g \in G$ and $\|\omega\|_{\ell_1(G)} = 1$. Therefore,

$$\widehat{\boldsymbol{f}}_{\mathrm{ERM}} = \big(\widehat{\boldsymbol{f}}_{\mathrm{ERM,\,triv}},\ \widehat{\boldsymbol{f}}_{\mathrm{ERM,\,non}}\big) \ \Longrightarrow \ \Pi\widehat{\boldsymbol{f}}_{\mathrm{ERM}} = \big(\widehat{\boldsymbol{f}}_{\mathrm{ERM,\,triv}},\ 0\big), \tag{A.63}$$

and, moreover,

$$\widehat{\boldsymbol{f}}_{\mathrm{ERM}} = \big(\widehat{\boldsymbol{f}}_{\mathrm{ERM,\,triv}},\ \widehat{\boldsymbol{f}}_{\mathrm{ERM,\,non}}\big) \ \Longrightarrow \ \mathbb{E}_\omega\widehat{\boldsymbol{f}}_{\mathrm{ERM}} = \big(\widehat{\boldsymbol{f}}_{\mathrm{ERM,\,triv}},\ \mathbb{E}_\omega'\,\widehat{\boldsymbol{f}}_{\mathrm{ERM,\,non}}\big), \tag{A.64}$$

where $\mathbb{E}_\omega'$ denotes the linear operator induced by $\mathbb{E}_\omega$ on $\mathcal{F}_G^\perp$. From the previous section, since $\mathbb{E}_\omega$ is $\epsilon$-weakly approximately symmetric, we have

$$\big\|\mathbb{E}_\omega'\widehat{\boldsymbol{f}}_{\mathrm{ERM}}\big\|_2^2 \ \le \ \epsilon\,\big\|\widehat{\boldsymbol{f}}_{\mathrm{ERM,\,non}}\big\|_2^2. \tag{A.65}$$

Therefore,

$$\mathcal{R}\left(\widehat{f}_{\mathrm{ERM}}^{\mathrm{wk}}\right) \;\leq\; \epsilon\,\mathbb{E}\left[\left\|\widehat{\boldsymbol{f}}_{\mathrm{ERM,\,non}}\right\|_2^2\right] \;+\; \Theta\left(\frac{\sigma^2 m_{\mathrm{triv}}}{n}\right). \tag{A.66}$$

Assuming $\|\boldsymbol{f}^\star\|_2^2 = \mathcal{O}(1)$, we obtain

$$\mathbb{E}\left[\left\|\widehat{\boldsymbol{f}}_{\mathrm{ERM,\,non}}\right\|_2^2\right] \;\leq\; 2\|\boldsymbol{f}^\star\|_2^2 \;+\; 2\,\mathcal{R}\left(\widehat{f}_{\mathrm{ERM}}\right) \;=\; \mathcal{O}(1). \tag{A.67}$$

Hence, the high-probability excess population risk under approximate symmetry satisfies

$$\mathcal{R}\left(\widehat{f}_{\mathrm{ERM}}^{\mathrm{wk}}\right) \;\leq\; \mathcal{O}(\epsilon) \;+\; \mathcal{O}\left(\frac{\sigma^2 m_{\mathrm{triv}}}{n}\right). \tag{A.68}$$

*Remark* 22. The three high-probability excess population risks derived in this subsection are

$$\mathcal{R}(\widehat{f}_{\mathrm{ERM}}) \;=\; \Theta\left(\frac{\sigma^2 m}{n}\right), \tag{A.69}$$

$$\mathcal{R}\left(\widehat{f}_{\mathrm{ERM}}^{\mathrm{ex}}\right) \;=\; \Theta\left(\frac{\sigma^2 m_{\mathrm{triv}}}{n}\right), \tag{A.70}$$

$$\mathcal{R}\left(\widehat{f}_{\mathrm{ERM}}^{\mathrm{wk}}\right) \;\leq\; \mathcal{O}\left(\frac{\sigma^2 m_{\mathrm{triv}}}{n}\right) \;+\; \mathcal{O}(\epsilon), \tag{A.71}$$

where $m = \dim(\mathcal{F})$ and $m_{\mathrm{triv}} = \dim(\mathcal{F}_G)$. Therefore, using Theorem 15, one can achieve the full generalization benefits of symmetry with an appropriate averaging scheme of size only $\mathcal{O}\left(\frac{\log|G|}{\epsilon}\right)$, without requiring $|G|$-fold averaging. Here $\epsilon$ can be chosen as the target generalization error. In particular, taking $\epsilon = \frac{\sigma^2 m_{\mathrm{triv}}}{n}$ makes the weakly symmetric estimator's generalization bound match (up to constants) the bound for exact symmetry enforcement (which is superior in this simple linear regression setting). The size of the averaging scheme is then only $\mathcal{O}\left(\frac{n\log|G|}{\sigma^2 m_{\mathrm{triv}}}\right)$, which can be much smaller than $|G|$.

# B  PROOF OF PROPOSITION 10

*Proof.* Note that the first two properties, as well as the last, follow directly from the definitions of averaging complexity for weak and strong approximate symmetry enforcement. In the second inequality, the universal upper bound $|G|$ on the averaging complexity follows from the uniform averaging scheme defined by

$$\omega(g) := \frac{1}{|G|}, \qquad \forall\, g \in G. \tag{B.1}$$

For this scheme, $\mathrm{size}(\omega) = |G|$, and for any $f \in \mathcal{F}$ we have

$$(\mathbb{E}_\omega[f])(x) \;=\; \frac{1}{|G|}\sum_{g\in G} f\left(g^{-1}x\right) \in \mathcal{F}_G, \tag{B.2}$$

which is exactly (and therefore also weakly and strongly approximately) symmetric, since it is the output of group averaging.

Moreover, we always have

$$\mathsf{AC}^{\mathrm{wk}}(\mathcal{F}, \varepsilon) \;\leq\; \mathsf{AC}^{\mathrm{st}}(\mathcal{F}, \varepsilon),$$

again by definition (similarly for other averaging complexities). Therefore, to complete the proof of Proposition 10, it suffices to establish the remaining inequality: for all $\varepsilon > 0$,

$$\mathsf{AC}^{\mathrm{st}}(\mathcal{F}, 4\varepsilon) \;\leq\; \mathsf{AC}^{\mathrm{wk}}(\mathcal{F}, \varepsilon). \tag{B.3}$$

To begin the proof, fix $\varepsilon > 0$ and let $\omega : G \to \mathbb{R}$ be an averaging scheme that attains $\mathsf{AC}^{\mathrm{wk}}(\mathcal{F}, \varepsilon)$. By definition, for all $f \in \mathcal{F}$,

$$\mathbb{E}_g\left[\left\|(\mathbb{E}_\omega[f])(x) - (\mathbb{E}_\omega[f])(gx)\right\|_{L^2(\mathcal{X})}^2\right] \;\leq\; \varepsilon\, \mathbb{E}_g\left[\|f(x) - f(gx)\|_{L^2(\mathcal{X})}^2\right].$$

We show that the same scheme $\omega$ achieves strong approximate symmetry with precision $4\varepsilon$.

Fix any $g' \in G$. By the triangle inequality and introducing the group-averaging operator $\mathbb{E}_g$ (uniform over $G$), we have

$$\|(\mathbb{E}_\omega[f])(x) - (\mathbb{E}_\omega[f])(g'x)\|_{L^2(\mathcal{X})} \leq \|(\mathbb{E}_\omega[f])(x) - \mathbb{E}_g[(\mathbb{E}_\omega[f])(gx)]\|_{L^2(\mathcal{X})}$$
$$+ \|\mathbb{E}_g[(\mathbb{E}_\omega[f])(gx)] - (\mathbb{E}_\omega[f])(g'x)\|_{L^2(\mathcal{X})}.$$

Since the group action on the domain preserves the measure $(d\mu(gx) = d\mu(x))$, we have

$$\|\mathbb{E}_g[(\mathbb{E}_\omega[f])(gx)] - (\mathbb{E}_\omega[f])(g'x)\|_{L^2(\mathcal{X})} = \|\mathbb{E}_g[(\mathbb{E}_\omega[f])(gg'^{-1}x)] - (\mathbb{E}_\omega[f])(x)\|_{L^2(\mathcal{X})} \quad \text{(B.4)}$$
$$= \mathbb{E}_g[(\mathbb{E}_\omega[f])(gx)] - (\mathbb{E}_\omega[f])(x)\|_{L^2(\mathcal{X})}. \quad \text{(B.5)}$$

Therefore, we have

$$\|(\mathbb{E}_\omega[f])(x) - (\mathbb{E}_\omega[f])(g'x)\|_{L^2(\mathcal{X})} \leq 2\,\|(\mathbb{E}_\omega[f])(x) - \mathbb{E}_g[(\mathbb{E}_\omega[f])(gx)]\|_{L^2(\mathcal{X})}.$$

By Jensen's inequality,

$$\|(\mathbb{E}_\omega[f])(x) - \mathbb{E}_g[(\mathbb{E}_\omega[f])(gx)]\|_{L^2(\mathcal{X})} \leq \mathbb{E}_g\big[\|(\mathbb{E}_\omega[f])(x) - (\mathbb{E}_\omega[f])(gx)\|_{L^2(\mathcal{X})}\big]$$
$$= \sqrt{\mathbb{E}_g\big[\|(\mathbb{E}_\omega[f])(x) - (\mathbb{E}_\omega[f])(gx)\|^2_{L^2(\mathcal{X})}\big]},$$

and therefore

$$\forall g' \in G: \quad \|(\mathbb{E}_\omega[f])(x) - (\mathbb{E}_\omega[f])(g'x)\|_{L^2(\mathcal{X})} \leq 2\sqrt{\varepsilon}\,\mathbb{E}_g\left[\|f(x) - f(gx)\|_{L^2(\mathcal{X})}\right].$$

Squaring both sides yields

$$\forall g' \in G: \quad \|(\mathbb{E}_\omega[f])(x) - (\mathbb{E}_\omega[f])(g'x)\|^2_{L^2(\mathcal{X})} = 4\varepsilon\,\left(\mathbb{E}_g\left[\|f(x) - f(gx)\|_{L^2(\mathcal{X})}\right]\right)^2 \quad \text{(B.6)}$$
$$\leq 4\varepsilon\,\mathbb{E}_g\left[\|f(x) - f(gx)\|^2_{L^2(\mathcal{X})}\right], \quad \text{(B.7)}$$

where we used the Cauchy–Schwarz inequality in the last step. Thus, the same averaging scheme (with the same size) achieves strong approximate symmetry with precision $4\varepsilon$, which completes the proof in the sense of the definition of the averaging complexity of the strong approximate symmetry enforcement. $\qquad\square$

*Remark* 23. Proposition 10 allows us to focus on weak approximate symmetry enforcement: the strong notion follows with only a constant-factor loss in precision: for all $\varepsilon > 0$, $\mathsf{AC}^{\mathrm{st}}(\mathcal{F}, 4\varepsilon) \leq \mathsf{AC}^{\mathrm{wk}}(\mathcal{F}, \varepsilon)$. Consequently, the upper bound we prove, $\Theta\big(\log|G|/\varepsilon\big)$, holds up to constants for both notions. From a theoretical perspective, this is significant because it lets one upgrade average-case error over the group to a uniform (worst-case) guarantee over all $g \in G$ within a constant factor.

Finally, we note that an analogous constant-factor relationship between uniform and average-case errors has recently been observed in the problem of testing symmetries in data; see (Soleymani et al., 2025b) for details.

## C  Proof of Theorem 13

*Proof.* We begin by recalling why it suffices to prove the bound on the averaging complexity for $\mathsf{K} = \min\left\{|G|, \sum_{\lambda \in \Lambda} M_\lambda - 1\right\}$, By the definition of tensor powers, the space $\widetilde{\mathrm{Sym}}^{\otimes k}(\mathcal{F}) = \bigoplus_{\ell=0}^{k} \mathrm{Sym}^{\otimes \ell}(\mathcal{F})$ is the direct sum of tensor product spaces of degrees $\ell = 0, 1, \ldots, k$. Consequently, for any $k' \geq k$ we have

$$\widetilde{\mathrm{Sym}}^{\otimes k}(\mathcal{F}) = \bigoplus_{\ell=0}^{k} \mathrm{Sym}^{\otimes \ell}(\mathcal{F}) \subseteq \bigoplus_{\ell=0}^{k'} \mathrm{Sym}^{\otimes \ell}(\mathcal{F}) = \widetilde{\mathrm{Sym}}^{\otimes k'}(\mathcal{F}), \quad \text{(C.1)}$$

where the inclusion follows from the monotonicity of direct sums of vector spaces.

According to Proposition 10, the averaging complexity of exact symmetry enforcement is monotone with respect to inclusion of vector spaces: if $\mathcal{F}_1 \subseteq \mathcal{F}_2$, then $\mathsf{AC}^{\mathrm{ex}}(\mathcal{F}_1) \leq \mathsf{AC}^{\mathrm{ex}}(\mathcal{F}_2)$. Specializing this inequality to $\mathcal{F}_1 = \widetilde{\mathrm{Sym}}^{\otimes \mathsf{K}}(\mathcal{F})$ and $\mathcal{F}_2 = \widetilde{\mathrm{Sym}}^{\otimes k}(\mathcal{F})$ for $k \geq \mathsf{K}$, we obtain

$$\mathsf{AC}^{\mathrm{ex}}(\widetilde{\mathrm{Sym}}^{\otimes \mathsf{K}}(\mathcal{F})) \leq \mathsf{AC}^{\mathrm{ex}}(\widetilde{\mathrm{Sym}}^{\otimes k}(\mathcal{F})), \quad \forall k \geq \mathsf{K}. \quad \text{(C.2)}$$

Moreover, Proposition 10 also implies that $\mathrm{AC}^{\mathrm{ex}}(\widetilde{\mathrm{Sym}}^{\otimes k}(\mathcal{F})) \leq |G|$ for all $k \in \mathbb{N}$. Therefore, to prove Theorem 13, it suffices to establish that

$$\mathrm{AC}^{\mathrm{ex}}(\widetilde{\mathrm{Sym}}^{\otimes \mathsf{K}}(\mathcal{F})) = |G|, \quad \text{where} \quad \mathsf{K} = \min\left\{|G|, \sum_{\lambda \in \Lambda} M_\lambda - 1\right\}.$$

We complete the proof of Theorem 13 through the following two claims, whose proofs are deferred to the end of this section. For background material required in these arguments, we refer the reader to Appendix A.

**Claim 24** (Steinberg (2014); Kollár & Tiep (2024)). *Let $\pi_i$, $i \in [r]$, $r = |\widehat{G}|$, denote all the irreducible representations of a finite group $G$. Consider the decomposition of the action of $G$ on the function space $\mathcal{F}$ (which we have already assumed to be faithful):*

$$\rho \cong \bigoplus_{i \in [r]} m_i\, \pi_i, \qquad m_i \in \mathbb{Z}_{\geq 0}. \tag{C.3}$$

*Define $\mathsf{K} = \min\left\{|G|, \sum_{\lambda \in \Lambda} M_\lambda - 1\right\}$. Moreover, for each $k \in [\mathsf{K}]$, decompose the induced representation of $G$ on the tensor power as*

$$\mathrm{Sym}^{\otimes k}(\rho) \cong \bigoplus_{i=1}^{|\widehat{G}|} m_i^{(k)}\, \pi_i, \qquad m_i^{(k)} \in \mathbb{Z}_{\geq 0}. \tag{C.4}$$

*Then, we have*

$$\forall i \in [r], \quad \exists k \in [\mathsf{K}] \text{ such that } m_i^{(k)} \geq 1. \tag{C.5}$$

Claim 24 shows that by taking tensor powers up to order $\mathsf{K} = \min\left\{|G|, \sum_{\lambda \in \Lambda} M_\lambda - 1\right\}$, we "observe" every irreducible representation at least once among the decompositions of the tensor powers. Indeed, we have

$$\widetilde{\mathrm{Sym}}^{\otimes \mathsf{K}}(\mathcal{F}) = \bigoplus_{k=0}^{\mathsf{K}} \mathrm{Sym}^{\otimes k}(\mathcal{F}) \implies \widetilde{\mathrm{Sym}}^{\otimes \mathsf{K}}(\rho) \cong \bigoplus_{k=0}^{\mathsf{K}} \mathrm{Sym}^{\otimes k}(\rho) \cong \bigoplus_{i=1}^{|\widehat{G}|} \underbrace{\left(\sum_{k=0}^{\mathsf{K}} m_i^{(k)}\right)}_{\geq 1 \text{ by Claim } 24} \pi_i.$$

In other words, the induced group action on $\widetilde{\mathrm{Sym}}^{\otimes \mathsf{K}}(\mathcal{F})$, denoted by $\widetilde{\mathrm{Sym}}^{\otimes \mathsf{K}(\rho)}$, is the direct sum of the representations on all tensor powers up to order $\mathsf{K}$. Furthermore, in the decomposition of $\widetilde{\mathrm{Sym}}^{\otimes \mathsf{K}}(\rho)$, every irreducible representation appears at least once. We will use this fact to establish lower bounds on averaging complexity via Fourier analysis on finite groups.

Let us now present the final claim needed to complete the proof.

**Claim 25.** *Consider an averaging scheme $\omega : G \to \mathbb{R}$ that achieves exact symmetry on the function space $\mathcal{F}$ with induced representation $\rho$. Assume that the decomposition of $\rho$ into irreducible representations of the finite group $G$ satisfies*

$$\rho \cong \bigoplus_{i \in [r]} m_i\, \pi_i, \qquad m_i \in \mathbb{Z}_{\geq 1}. \tag{C.6}$$

*Then, we have*

$$\sum_{g \in G} \omega(g)\, \pi(g)^* = 0 \in \mathbb{R}^{d_\pi \times d_\pi}, \tag{C.7}$$

*for all nontrivial irreducible representations $\pi \in \widehat{G}$, where $\widehat{G}$ denotes the set of all irreducible representations of $G$.*

By Claim 25, the Fourier transform of the *group signal* $\omega : G \to \mathbb{R}$ is *sparse*, in the sense that

$$\widehat{\omega}(\pi) := \sum_{g \in G} \omega(g)\, \pi(g)^* = 0 \in \mathbb{R}^{d_\pi \times d_\pi}, \tag{C.8}$$

for every non-trivial irrep $\pi \in \widehat{G}$.

Moreover, the conditions of Claim 25 are already satisfied by the representation $\widetilde{\mathrm{Sym}}^{\otimes \mathsf{K}}(\rho)$ induced on $\widetilde{\mathrm{Sym}}^{\otimes \mathsf{K}}(\mathcal{F})$, thanks to Claim 24. Therefore, combining the two claims and applying the Fourier inversion formula for the group signal $\omega$, we conclude that if $\omega$ achieves exact symmetry for the function class, then necessarily

$$\widehat{\omega}(\pi) = 0 \quad \forall \pi \text{ non-trivial} \quad \Longrightarrow \quad \omega(g) = \frac{1}{|G|}, \quad \forall g \in G \quad \Longrightarrow \quad \mathrm{size}(\omega) = |G|. \tag{C.9}$$

Here we used the fact that a group signal with Fourier support only on the trivial irrep must be constant, along with the assumption that $\|\omega\|_{\ell_1(G)} = \sum_{g \in G} \omega(g) = 1$. For further details on Fourier analysis on finite groups, see Appendix A.5.

This completes the proof of Theorem 13. In the remainder of this section, we provide the proofs of the claims stated above.

$\square$

## C.1 PROOF OF CLAIM 24

*Proof.* To prove the result, first note that the claim $\mathsf{K} = |G|$ follows directly from recent work of Kollár & Tiep (2024). The remainder of the proof draws substantial inspiration from Steinberg (2014).

For each irreducible representation $\pi_i$, define

$$F_i(t) := \sum_{k \geq 0} m_i^{(k)} t^k, \qquad m_i^{(k)} := [\pi_i : \mathrm{Sym}^{\otimes k}(\rho)].$$

By Molien's formula,

$$F_i(t) = \frac{1}{|G|} \sum_{g \in G} \frac{\chi_i(g^{-1})}{\det(I - t\,\rho(g))}.$$

Now let

$$D(t) := \prod_{\lambda \in \Lambda} (1 - \lambda t)^{M_\lambda}, \qquad M_\lambda := \max_{g \in G} \mathrm{mult}_\lambda(\rho(g)).$$

Since $\det(I - t\,\rho(g)) \mid D(t)$ for every $g \in G$, we may write

$$F_i(t) = \frac{P_i(t)}{D(t)}$$

for some polynomial $P_i$ with

$$\deg P_i < \deg D = \sum_{\lambda \in \Lambda} M_\lambda.$$

If

$$m_i^{(0)} = \cdots = m_i^{\left(\sum_{\lambda \in \Lambda} M_\lambda - 1\right)} = 0,$$

then $F_i(t)$ vanishes to order at least $\deg D$ at $t = 0$, hence so does $P_i(t) = D(t) F_i(t)$. Since $\deg P_i < \deg D$, this forces $P_i \equiv 0$, i.e. $F_i \equiv 0$.

It remains to show that $F_i \not\equiv 0$. The identity element contributes

$$\frac{1}{|G|} \frac{\chi_i(e)}{(1-t)^{\dim \mathcal{F}}} = \frac{1}{|G|} \frac{\dim(\pi_i)}{(1-t)^{\dim \mathcal{F}}},$$

which has a pole of order $\dim \mathcal{F}$ at $t = 1$. For $g \neq e$, faithfulness of $\rho$ implies $\rho(g) \neq I$, so the multiplicity of the eigenvalue 1 in $\rho(g)$ is at most $\dim \mathcal{F} - 1$. Hence every nonidentity summand

has pole order at most $\dim \mathcal{F} - 1$ at $t = 1$. Therefore the coefficient of $(1 - t)^{-\dim \mathcal{F}}$ in $F_i(t)$ is $\dim(\pi_i)/|G| \neq 0$, and so $F_i \not\equiv 0$.

Consequently, for every $i$, there exists

$$k \in \left\{0, 1, \ldots, \sum_{\lambda \in \Lambda} M_\lambda - 1\right\}$$

such that $m_i^{(k)} \geq 1$. $\qquad \square$

## C.2 PROOF OF CLAIM 25

*Proof.* Throughout the proof, we adopt the notation and definitions from Appendix A, in particular those introduced in Appendix A.6. Let $\omega : G \to \mathbb{R}$ denote an averaging scheme, and let $\rho$ be the representation induced on the function class $\mathcal{F}$, decomposed into irreps as

$$\rho \cong \bigoplus_{i \in [r]} m_i \, \pi_i, \qquad m_i \in \mathbb{Z}_{\geq 1}, \tag{C.10}$$

where $r := |\widehat{G}|$ denotes the number of distinct irreps.

Our goal is to show that, under the condition $m_i \geq 1$ for all $i$, and assuming that $\omega$ is an exactly symmetric averaging scheme, the nontrivial components of the Fourier transform of $\omega$ vanish:

$$\sum_{g \in G} \omega(g) \, \pi(g)^* \; = \; 0 \in \mathbb{R}^{d_\pi \times d_\pi}, \tag{C.11}$$

for all nontrivial irreducible representations $\pi \in \widehat{G}$, where $\widehat{G}$ denotes the set of irreducible representations of $G$.

Note that, after a change of coordinates (i.e., choosing an appropriate basis), we can write the group representation $\rho$ in block-diagonal form:

$$\rho(g) \; = \; \bigoplus_{i \in [r]} \big(I_{m_i} \otimes \pi_i(g)\big) \; \in \; \mathbb{R}^{m \times m}, \qquad \forall g \in G, \tag{C.12}$$

where $I_{m_i} \in \mathbb{R}^{m_i \times m_i}$ denotes the identity matrix for each $i \in [r]$, and

$$m \; := \; \dim(\mathcal{F}) \; = \; \sum_{i=1}^{r} m_i d_{\pi_i},$$

with $d_{\pi_i} = \dim(\pi_i)$.

Therefore, there exist projection matrices $\Pi_i \in \mathbb{C}^{m \times m}$, one for each $i \in [r]$, corresponding to the subspaces spanned by the (possibly multiple) copies of $\pi_i$. In the chosen coordinates, each projection takes the form

$$\Pi_i \; = \; 0 \oplus 0 \oplus \cdots \oplus 0 \oplus I_{m_i d_{\pi_i}} \oplus 0 \oplus \cdots \oplus 0, \qquad \forall i \in [r]. \tag{C.13}$$

In the orthonormal basis of the function space $\mathcal{F}$, any function $f \in \mathcal{F}$ can be identified with its coefficient vector $\boldsymbol{f} \in \mathbb{C}^m$. We decompose this vector as

$$\boldsymbol{f} \; = \; (\boldsymbol{f}_1, \, \boldsymbol{f}_2, \, \ldots, \, \boldsymbol{f}_r),$$

where each block $\boldsymbol{f}_i$ corresponds to the component associated with $\pi_i$.

By definition of the trivial representation (assumed here to be indexed by $i = 1$), we have

$$f \in \mathcal{F}_G \quad \Longleftrightarrow \quad \boldsymbol{f} \; = \; (\boldsymbol{f}_1, \boldsymbol{0}, \boldsymbol{0}, \, \ldots, \, \boldsymbol{0}) \in \mathbb{C}^m, \tag{C.14}$$

so that $\Pi_1$ is precisely the projection onto the subspace of exactly symmetric functions, i.e. $\mathcal{F}_G \subseteq \mathcal{F}$.

Note that a given averaging scheme $\omega : G \to \mathbb{R}$ induces a linear operator $\mathbb{E}_\omega : \mathcal{F} \to \mathcal{F}$. With a slight abuse of notation, we use the same symbol for its action on coefficient vectors, so that we may also regard $\mathbb{E}_\omega : \mathbb{C}^m \to \mathbb{C}^m$.

Since $\omega$ enforces exact symmetry, we must have

$$\mathbb{E}_\omega \boldsymbol{f} = \left( \star, \, \boldsymbol{0}, \, \boldsymbol{0}, \dots \right) \in \mathbb{C}^m, \qquad \forall \boldsymbol{f} \in \mathbb{C}^m. \tag{C.15}$$

In other words, because the output of the averaging operator is exactly symmetric, all components corresponding to nontrivial irreps must vanish in the coefficient vector.

Now for arbitrary $\boldsymbol{f} \in \mathbb{C}^m$, we have

$$\mathbb{E}_\omega \boldsymbol{f} = \sum_{g \in G} \omega(g) \, \rho(g) \boldsymbol{f} = \left( \star, \, \boldsymbol{0}, \, \boldsymbol{0}, \dots \right) \in \mathbb{C}^m. \tag{C.16}$$

Therefore, for any $i \geq 2$ (indices corresponding to nontrivial irreps), applying the projection matrix $\Pi_i$ to the above identity yields

$$\Pi_i \sum_{g \in G} \omega(g) \, \rho(g) \boldsymbol{f} = \sum_{g \in G} \omega(g) \, \Pi_i \rho(g) \boldsymbol{f} \tag{C.17}$$

$$= \sum_{g \in G} \omega(g) \, \pi_i(g)^{\oplus m_i} \, \boldsymbol{f}_i \tag{C.18}$$

$$= \left( \sum_{g \in G} \omega(g) \, \pi_i(g) \right)^{\oplus m_i} \boldsymbol{f}_i \tag{C.19}$$

$$= \boldsymbol{0} \in \mathbb{C}^m. \tag{C.20}$$

This identity must hold for all $\boldsymbol{f}_i \in \mathbb{C}^{m_i}$, and since $m_i \geq 1$ by assumption, we conclude that

$$\sum_{g \in G} \omega(g) \, \pi_i(g) = \boldsymbol{0} \in \mathbb{C}^{d_{\pi_i} \times d_{\pi_i}}, \qquad \forall \, i \geq 2. \tag{C.21}$$

Taking the conjugate transpose of the above identity completes the proof.

$\square$

## D  PROOF OF THEOREM 15

*Proof.* Throughout the proof, we rely on the tools and ideas developed in Appendix A, as well as those used in the proof of Theorem 13. We briefly review them here.

Let $\mathcal{F}$ denote an arbitrary function class, and let $\rho$ be the representation induced by the action of the finite group $G$ on $\mathcal{F}$, which decomposes into irreducibles as

$$\rho \cong \bigoplus_{i \in [r]} m_i \, \pi_i, \qquad m_i \in \mathbb{Z}_{\geq 0}, \tag{D.1}$$

where $r := |\widehat{G}|$ is the number of distinct irreps. Note that $m_i$ may be zero for some indices $i$.

Under a change of coordinates (i.e., after choosing an appropriate basis for $\mathcal{F}$), the group representation $\rho$ can be expressed in block-diagonal form:

$$\rho(g) = \bigoplus_{i \in [r]} \left( I_{m_i} \otimes \pi_i(g) \right) \in \mathbb{R}^{m \times m}, \qquad g \in G, \tag{D.2}$$

where $I_{m_i} \in \mathbb{R}^{m_i \times m_i}$ denotes the identity matrix for each $i \in [r]$. Here

$$m := \dim(\mathcal{F}) = \sum_{i=1}^{r} m_i \, d_{\pi_i}, \qquad d_{\pi_i} = \dim(\pi_i).$$

Therefore, there exist projection matrices $\Pi_i \in \mathbb{C}^{m \times m}$, one for each $i \in [r]$, corresponding to the subspaces spanned by the (possibly multiple, or zero) copies of $\pi_i$. In the chosen coordinates, each projection has the form

$$\Pi_i = 0 \oplus 0 \oplus \cdots \oplus 0 \oplus I_{m_i d_{\pi_i}} \oplus 0 \oplus \cdots \oplus 0, \qquad i \in [r], \tag{D.3}$$

where the identity block appears in the position associated with $\pi_i$.

In the orthonormal basis of the function space $\mathcal{F}$, any function $f \in \mathcal{F}$ can be identified with its coefficient vector $\boldsymbol{f} \in \mathbb{C}^m$. We decompose this vector as

$$\boldsymbol{f} = (\boldsymbol{f}_1, \boldsymbol{f}_2, \ldots, \boldsymbol{f}_r),$$

where each block $\boldsymbol{f}_i \in \mathbb{C}^{m_i d_{\pi_i}}$ corresponds to the isotypic component associated with $\pi_i$.

By convention, we assume the trivial representation is indexed by $i = 1$. Then

$$f \in \mathcal{F}_G \quad \Longleftrightarrow \quad \boldsymbol{f} = (\boldsymbol{f}_1, \boldsymbol{0}, \boldsymbol{0}, \ldots, \boldsymbol{0}) \in \mathbb{C}^m, \tag{D.4}$$

so that $\Pi_1$ is exactly the projection onto the subspace of symmetric functions, i.e., $\mathcal{F}_G \subseteq \mathcal{F}$.

Note that, according to Proposition 10, it suffices to prove Theorem 15 for weak approximate symmetry. Indeed, once the claim is established in the weak case, we have

$$\mathsf{AC}^{\mathrm{st}}(\mathcal{F}, \varepsilon) \leq \mathsf{AC}^{\mathrm{wk}}(\mathcal{F}, \varepsilon/4) = \mathcal{O}\left(\frac{\log|G|}{\varepsilon}\right). \tag{D.5}$$

Therefore, throughout this section we focus only on weak approximate symmetry enforcement.

Consider an averaging scheme $\omega : G \to \mathbb{R}$ that induces a linear operator $\mathbb{E}_\omega : \mathcal{F} \to \mathcal{F}$. With a slight abuse of notation, we use the same symbol for its action on coefficient vectors, so that we may also regard $\mathbb{E}_\omega : \mathbb{C}^m \to \mathbb{C}^m$. Assume that $\mathbb{E}_\omega[\cdot]$ enforces $\epsilon$-weak approximate symmetry for a fixed parameter $\epsilon > 0$.

As noted at the end of Appendix A.7, this condition can be written as

$$\mathbb{E}_\omega[\cdot] \text{ is } \epsilon\text{-weakly symmetric} \quad \Longleftrightarrow \quad \left\|\sum_{i=2}^r \Pi_i \mathbb{E}_\omega \boldsymbol{f}\right\|_2^2 \leq \epsilon \left\|\sum_{i=2}^r \Pi_i \boldsymbol{f}\right\|_2^2, \quad \forall f \in \mathcal{F}. \tag{D.6}$$

Equivalently,

$$\mathbb{E}_\omega[\cdot] \text{ is } \epsilon\text{-weakly symmetric} \quad \Longleftrightarrow \quad \sum_{i=2}^r \|\Pi_i \mathbb{E}_\omega \boldsymbol{f}\|_2^2 \leq \epsilon \sum_{i=2}^r \|\Pi_i \boldsymbol{f}\|_2^2, \quad \forall f \in \mathcal{F}. \tag{D.7}$$

A necessary and sufficient condition for the above inequality is to require that

$$\forall i \geq 2 : \quad \|\Pi_i \mathbb{E}_\omega \boldsymbol{f}\|_2^2 \leq \epsilon \|\Pi_i \boldsymbol{f}\|_2^2, \quad \forall f \in \mathcal{F}. \tag{D.8}$$

Using the decomposition of the representation $\rho$ into irreps, this condition reduces to

$$\forall i \geq 2 : \quad \left\|\sum_{g \in G} \omega(g) \, \pi_i(g)^{\oplus m_i} \, \Pi_i \boldsymbol{f}\right\|_2^2 \leq \epsilon \|\Pi_i \boldsymbol{f}\|_2^2, \quad \forall f \in \mathcal{F}. \tag{D.9}$$

A necessary and sufficient condition for this to hold is

$$\sup_{i \geq 2} \left\|\sum_{g \in G} \omega(g) \, \pi_i(g)\right\|_{\mathrm{op}}^2 \leq \epsilon. \tag{D.10}$$

Let us now use a probabilistic construction for $\omega : G \to \mathbb{R}$. Draw $n$ i.i.d. samples uniformly from $G$, and let $\Omega$ denote the empirical measure induced by these $n$ samples. We use the capital letter $\Omega$ instead of $\omega$ to emphasize that it is constructed randomly.

Since each $\pi_i$ is a nontrivial irrep, we have

$$\mathbb{E}_g[\pi_i(g)] = 0 \in \mathbb{C}^{d_{\pi_i} \times d_{\pi_i}}, \qquad \forall i \geq 2. \tag{D.11}$$

Moreover, since all representations considered in this paper are unitary, it follows that

$$\sup_{i \geq 2} \sup_{g \in G} \|\pi_i(g)\|_{\mathrm{op}} \leq 1. \tag{D.12}$$

Now we apply the matrix Bernstein tail bound from Tropp (2012, Theorem 1.6). In their notation, we have $R = 1$, $\sigma^2 \leq n$, and $t^2 = n^2\epsilon$. Then, for any $\epsilon < 1$, we obtain

$$\mathbb{P}_\Omega \left( \left\| \sum_{g \in G} \Omega(g)\pi_i(g) \right\|_{\mathrm{op}}^2 > \epsilon \right) \leq 2d_{\pi_i} \exp\left(-\tfrac{3n\epsilon}{8}\right), \qquad \forall i \geq 2. \tag{D.13}$$

Applying a union bound then gives

$$\mathbb{P}_\Omega \left( \sup_{i \geq 2} \left\| \sum_{g \in G} \Omega(g)\pi_i(g) \right\|_{\mathrm{op}}^2 > \epsilon \right) \leq 2 \sum_{i \geq 2} d_{\pi_i} \exp\left(-\tfrac{3n\epsilon}{8}\right) \tag{D.14}$$

$$\leq 2|G| \exp\left(-\tfrac{3n\epsilon}{8}\right), \tag{D.15}$$

where in the last step we used the fact that

$$\sum_{i \geq 2} d_{\pi_i} \leq \sum_{i \geq 2} d_{\pi_i}^2 = |G| - 1. \tag{D.16}$$

Thus, to ensure that the probability of failure of a random averaging scheme to satisfy the weak approximate symmetry condition is at most $\delta < 1$, it suffices to take

$$n = \left\lceil 2.67 \times \frac{\log|G| + \log\frac{1}{\delta} + 0.7}{\epsilon} \right\rceil, \tag{D.17}$$

samples. At the same time, the size of such a random averaging scheme is

$$\mathrm{size}(\Omega) = n = \mathcal{O}\left( \frac{\log|G| + \log\frac{1}{\delta}}{\epsilon} \right), \tag{D.18}$$

which completes the proof.

$\square$

*Remark* 26. In the proof, the decomposition into irreps and the removal of redundancies (i.e., cases with $m_i \geq 2$) are essential for obtaining the $\log|G|$ term. A naive application of matrix concentration inequalities to the entire space $\mathcal{F}$ would yield only a bound depending on $\log\dim(\mathcal{F})$, which can be suboptimal when the function space $\mathcal{F}$ is large. By contrast, through representation-theoretic arguments we derive a bound of $\log|G|$, which holds uniformly for *any* finite-dimensional function space $\mathcal{F}$.

*Remark* 27. The proofs of our main results on exactly symmetric functions are closely related to classical work in representation theory, including the results of Burnside (Burnside, 2012), Steinberg (Steinberg, 1962), and Brauer (Brauer, 1964).

The theory of designing averaging schemes is also closely connected to the study of unitary designs and unitary codes, which have been investigated in the literature (Roy & Scott, 2009; Dankert et al., 2009). The notion of almost independent permutations is also closely related to our setting, in the specific case of the symmetric group and low-degree polynomials (Alon & Lovett, 2013). Moreover, the fact that under a random averaging scheme logarithmically sized subsets of group elements suffice to ensure that all nontrivial irreps average close to zero has been used in a different context in the study of random walks on groups (see the Alon–Roichman theorem (Alon & Roichman, 1994)). This line of work is further related to the theory of Cayley graphs and expander graphs (Bourgain & Gamburd, 2008), as well as tensor product Markov chains (Benkart et al., 2020), both of which have numerous applications (Hoory et al., 2006).

## E   PROOF OF THE CLAIM IN REMARK 16

*Proof.* In order to show that at least $\Omega_\varepsilon\big(\log|G|\big)$ action queries (AQs) are required to achieve approximate symmetry, we construct a particular instance of the problem.

Assume that $\epsilon < 1$, and let us consider the group $G = \{0,1\}^d$ under addition modulo two, where $d \in \mathbb{N}$. Note that $\log |G| = \Theta(d)$. Let $\pi_i$, $i \in [r]$, denote the distinct irreps of $G$, which are all one-dimensional since $G$ is a commutative group. This means that $r = |G|$. Consider an arbitrary averaging scheme $\omega : G \to \mathbb{R}$ that achieves weak approximate symmetry (Definition 6).

Using the same line of argument as appeared in the proof of Theorem 15 (Equation D.10), we have

$$|\widehat{\omega}(\pi_i)|^2 = \left| \sum_{g \in G} \omega(g)\, \pi_i(g) \right|^2 \leq \epsilon, \quad \forall i : i \geq 2, \tag{E.1}$$

where $i = 1$ is used above to denote the trivial irrep.

We claim that this means that the support of $\omega$ is a generating set of the group. In other words, letting $S := \{\, g \in G : \omega(g) \neq 0 \,\}$ we claim that $S$ generates the group. This means that there exists a finite $k \in \mathbb{N}$ such that $\cup_{\ell \in [k]} S^\ell = G$, where we define $A^\ell := \{\sum_{j=1}^\ell a_i : a_i \in A, \forall i \in [\ell]\}$ for any set $A \subseteq G$.

First, let us show that the above claim completes the proof. Note that $G$ is a $d$-dimensional vector space, and thus if $S$ has fewer than $d$ elements then it is impossible to have $\cup_{\ell \in [k]} S^\ell = G$, via elementary linear algebra arguments (i.e., span of less than $d$ vectors cannot become a $d$-dimensional vector space). Indeed, in such cases we have $\cup_{\ell=1}^\infty S^\ell \subsetneq G$. This means that $|S| \geq d = \Theta(\log |G|)$. However, the size of the averaging scheme $\omega$ is $\mathrm{size}(\omega) = |S| = \Theta(\log |G|)$. Since this bound holds for all $\epsilon < 1$, the proof is complete.

Now let us focus on proving that such a subset $S$ generates the group. For any two functions $\omega_1, \omega_2 : G \to \mathbb{R}$, define their convolution, denoted by $\omega_1 \star \omega_2 : G \to \mathbb{R}$, such that

$$(\omega_1 \star \omega_2)(g) := \sum_{h \in G} \omega_1(h)\omega_2\left(h^{-1}g\right), \quad \forall g \in G. \tag{E.2}$$

A clear property of the convolution operator is that

$$\mathrm{supp}(\omega_1 \star \omega_2) \subseteq \mathrm{supp}(\omega_1) + \mathrm{supp}(\omega_2), \quad \text{for all } \omega_1, \omega_2. \tag{E.3}$$

In particular, this shows that for the averaging scheme $\omega : G \to \mathbb{R}$ and its $\ell$-fold convolution

$$\omega^{\star \ell} := \underbrace{\omega \star \omega \star \ldots \star \omega}_{\ell \text{ times}}, \quad \forall \ell \in \mathbb{N}, \tag{E.4}$$

we have

$$\mathrm{supp}(\omega^{\star \ell}) \subseteq (\mathrm{supp}(\omega))^\ell, \quad \forall \ell \in \mathbb{N}. \tag{E.5}$$

This means that

$$\bigcup_{\ell \in [k]} \mathrm{supp}\left(\omega^{\star \ell}\right) \subseteq \bigcup_{\ell \in [k]} (\mathrm{supp}(\omega))^\ell, \quad \forall k \in \mathbb{N}. \tag{E.6}$$

Therefore, to complete the proof, it is sufficient to show that

$$\bigcup_{\ell \in [k]} \mathrm{supp}\left(\omega^{\star \ell}\right) = G, \tag{E.7}$$

for some finite $k \in \mathbb{N}$.

Note that, according to the properties of the Fourier transform on groups, we have

$$\left|\widehat{\omega^{\star \ell}}(\pi_i)\right|^2 \leq |\widehat{\omega}(\pi_i)|^{2\ell} \leq \epsilon^\ell, \quad \forall i \geq 2. \tag{E.8}$$

In particular, since $\epsilon < 1$, we have that $\lim_{\ell \to \infty} \left|\widehat{\omega^{\star \ell}}(\pi_i)\right|^2 = 0$, uniformly over $i \geq 2$. This means that $\omega^{\star \ell}$ converges in $L^2(G)$ to the uniform distribution over $G$. Recall that $\omega$ is an averaging scheme, thus its average over the group is one. Moreover, convergence in $L^2(G)$ and pointwise convergence are essentially equivalent here, since $G$ is finite.

Therefore, since the support of the uniform distribution is the whole group, we conclude that for some finite $k \in \mathbb{N}$, we have that $\mathrm{supp}(\omega^{\star k}) = G$, and this completes the proof.

$\square$

*Remark* 28. The above lower bound indeed holds for all finite groups, if we replace $S$ with the minimum generating set of the group $G$. More precisely, the number of required action queries (AQs) is at least $\Omega_\epsilon(|S|)$, where $S$ here is the minimum-sized generating set of the group $G$. For the particular case with $G = \{0, 1\}^d$, we showed that any generating set has size at least $\log |G|$, thus proving the claim.

## F  Beyond $L^2(\mathcal{X})$: On Approximate Symmetry in Other Metrics

In this section, we discuss how choosing metrics other than the $L^2(\mathcal{X})$-distance can affect our results. Here, $\mathcal{X}$ is equipped with a Borel measure $\mu$, and $L^2(\mathcal{X})$ is defined with respect to $\mu$.

Assume that $\omega : G \to \mathbb{R}$ achieves weak (or strong) approximate symmetry with respect to a given parameter $\epsilon$. Let $f \in \mathcal{F} \subseteq L^2(\mathcal{X})$ be an arbitrary function. According to Definition 6, and using the characterization of weak approximate symmetry in Equation A.33, we have

$$\|\mathbb{E}_\omega[f] - \mathbb{E}_g[f]\|^2_{L^2(\mathcal{X})} \leq \epsilon \, \mathbb{E}_g[\|f(x) - f(gx)\|^2_{L^2(\mathcal{X})}] \tag{F.1}$$

$$\leq 4\epsilon \, \|f\|^2_{L^2(\mathcal{X})}. \tag{F.2}$$

In the above, the operator $\mathbb{E}_\omega[\cdot]$ is defined according to the averaging scheme, and $\mathbb{E}_g[\cdot]$ is the (full) group averaging operator corresponding to the uniform averaging scheme over the whole group.

The above inequality tells us that if we have a weak (or strong) averaging scheme, then the resulting averaged functions are $\epsilon$-close to their full group-averaged versions in the $L^2(\mathcal{X})$-metric. This holds for *all* square-integrable functions $f \in \mathcal{F}$. Moreover, the size of the averaging scheme is only $\text{size}(\omega) = \mathcal{O}\left(\frac{\log |G|}{\epsilon}\right)$, according to our main result in the paper.

What happens if we want to go beyond the $L^2(\mathcal{X})$-norm and provide approximations of group averaging using sparse sets? Let us discuss what happens if we want to achieve this for the supremum norm over $\mathcal{X}$ and $\mathcal{F}$ via a random averaging scheme $\omega : G \to \mathbb{R}$ derived by sampling uniformly from the group. This is motivated by a number of previous studies on approximate symmetry (Ashman et al., 2024; Kim et al., 2023).

For a fixed function $f \in \mathcal{F}$ and a fixed $x \in \mathcal{X}$, note that according to classical concentration inequalities (e.g., Hoeffding's inequality) we have

$$|(\mathbb{E}_\omega[f])(x) - (\mathbb{E}_g[f])(x)|^2 \leq \mathcal{O}\left(\frac{\|f\|^2_{L^\infty(\mathcal{X})} \log(\frac{1}{\delta})}{\text{size}(\omega)}\right), \quad \text{with probability } 1 - \delta. \tag{F.3}$$

This bound, while even independent of the group size, is less interesting since it only holds for one particular pair $(x, f)$. To make it more general, one may want to take a supremum (over $x$ and/or $f$) of the left-hand side of the above inequality and hope that a modified upper bound holds.

To take the supremum over $x \in \mathcal{X}$, we need to use the so-called *covering* arguments, which are standard in classical statistics. Let $\log \mathcal{N}(\kappa, \mathcal{X})$ denote the metric entropy of $\mathcal{X}$ at scale $\kappa$, that is, the logarithm of the minimum number of points required to cover the whole domain $\mathcal{X}$ with balls of radius at most $\kappa$, where we equip the domain with a given metric. Assume that $f \in L^2(\mathcal{X})$ is 1-Lipschitz with respect to the given metric.

Assume $\kappa^2 = \epsilon$ and

$$\text{size}(\omega) = \Omega\left(\frac{\|f\|^2_{L^\infty(\mathcal{X})} \log(\frac{1}{\delta}) + \|f\|^2_{L^\infty(\mathcal{X})} \log \mathcal{N}(\kappa, \mathcal{X})}{\epsilon}\right). \tag{F.4}$$

Then we have

$$\sup_{x \in \mathcal{X}} |(\mathbb{E}_\omega[f])(x) - (\mathbb{E}_g[f])(x)|^2 = \mathcal{O}(\epsilon), \quad \text{with probability } 1 - \delta, \tag{F.5}$$

Usually, the metric entropy depends linearly on the intrinsic dimension of the domain $\mathcal{X}$, and it is also heavily affected by the volume of the domain. The above bound, while being nice and independent of the group, still depends on potentially complicated constants determined by the geometry of the input domain $\mathcal{X}$.

There is one more issue here. The bound above holds only for a fixed function $f \in \mathcal{F}$. To obtain a uniform bound holding for all $f \in \mathcal{F}$, one needs to study covering numbers of the function space $\mathcal{F}$, which can be difficult to handle for general spaces.

Let us now obtain a uniform bound over functions $f \in \mathcal{F}$ to see how complicated this task can become. Consider a fixed $x \in \mathcal{X}$, and let $\mu_\omega$ and $\mu_g$ denote the probability measures corresponding to the law of the point $x$ transformed either according to the distribution $\omega$ or uniformly over the domain. Note that

$$|(\mathbb{E}_\omega[f])(x) - (\mathbb{E}_g[f])(x)| \leq \mathrm{Lip}(f)\, W(\mu_\omega, \mu_g), \tag{F.6}$$

for all $f \in \mathcal{F}$, where $W(\cdot, \cdot)$ denotes the (Wasserstein-1) optimal transport distance between measures on $\mathcal{X}$. This bound is indeed optimal whenever $\mathcal{F}$ contains all Lipschitz functions over $\mathcal{X}$. Let $\mathcal{F}_{\mathrm{Lip}}$ denote the set of all $L$-Lipschitz functions over $\mathcal{X}$, for some fixed $L \in \mathbb{R}$, and assume that $\mathcal{F} = \mathcal{F}_{\mathrm{Lip}}$. The empirical measure $\mu_\omega$ convergences in Wasserstein distance to $\mu_g$ as

$$\sup_{f \in \mathcal{F}_{\mathrm{Lip}}} |(\mathbb{E}_\omega[f])(x) - (\mathbb{E}_g[f])(x)| = L\, W(\mu_\omega, \mu_g), \quad \mathbb{E}[W(\mu_\omega, \mu_g)] \lesssim \sqrt{\frac{|G|}{\mathrm{size}(\omega)}}, \tag{F.7}$$

where the latter expectation is over the randomness of choosing $\omega$. Moreover, this upper bound is minimax optimal for general finite groups and data domains; see, for instance, (Han et al., 2015).

Note that there is a curse of dimensionality here: to ensure a bounded error, one needs averaging schemes of size at least $\mathrm{size}(\omega) = \Theta(|G|)$, which is often $\exp(\Theta(d))$. This is in contrast to the logarithmic bound in the group size for the $L^2(\mathcal{X})$-distance, which holds with no curse of dimensionality. As a final remark, note that all the analysis above holds only for a fixed $x \in \mathcal{X}$, and obtaining a uniform bound over $x \in \mathcal{X}$ introduces another layer of complexity.

To conclude, obtaining the same type of result uniformly over all $x \in \mathcal{X}$ and $f \in \mathcal{F}$ is impossible in full generality, even for Lipschitz function classes, which are a substantially smaller subclass of square-integrable functions. Moreover, since generalization analyses in machine learning and statistics are almost always governed by the $L^2(\mathcal{X})$-distance, going beyond this regime has less theoretical motivation; see Appendix A.8. Still, the problem of finding better bounds beyond the $L^2(\mathcal{X})$ regime for specific function classes $\mathcal{F}$ is an open direction that we leave for future work.

## G    EXPERIMENTS

In this section, we present a simple proof-of-concept experiment that validates the theoretical findings of this paper. We consider $n_{\mathrm{train}} = 5 \times 10^4$ training and $n_{\mathrm{test}} = 5 \times 10^4$ test samples in dimension $d = 20$. Each data point $x \in \mathbb{R}^d$ is drawn i.i.d. from a Gaussian distribution with zero mean and identity covariance, and is labeled according to the target regression function:

$$f^\star(x) \coloneqq \langle w^\star, \mathrm{abs}(x) \rangle,$$

where $\mathrm{abs}(x) \in \mathbb{R}^d$ denotes the element-wise absolute value of $x$, and $w^\star \in \mathbb{R}^d$ is an unknown weight vector sampled from a zero-mean Gaussian with identity covariance.

To learn $f^\star$, we train a three-layer ReLU network with two hidden layers of widths $h_1 = 128$ and $h_2 = 64$. The network is trained using SGD with learning rate $10^{-3}$ and batch size 256 for 500 epochs, using the squared loss.

By construction, this task is invariant under coordinate-wise sign flips, meaning that for any $g \in G \coloneqq \{\pm 1\}^d$, we have $f^\star(gx) = f^\star(x)$. The group $G$ therefore has cardinality $|G| = 2^d$, which is prohibitively large for exact group averaging in practice. To approximate group averaging, we instead sample a random subset $S \subset G$ of size $|S| = 2^k$ for $k \in \{0, 1, \ldots, 10\}$. At evaluation time, the prediction on an input $x$ is obtained by averaging the network outputs over all transformations in $S$, and the test loss is computed via the squared loss. Crucially, the subset $S$ is fixed throughout training and is used *only* at evaluation time, and the training procedure itself does not depend on $S$.

Figure 2 summarizes the results of this experiment. The left plot shows the final test loss as a function of the subset size $|S|$. As $|S|$ increases, the test loss decreases, reflecting the benefit of averaging over more group elements. Interestingly, most of the improvement is already achieved around $|S| = 32$, and larger subsets yield only marginal gains. This behavior is in strong agreement

with our theory, which predicts that logarithmic-sized subsets already capture essentially the full benefit of group averaging.

The right plot in Figure 2 illustrates how averaging with a subset of size $|S| = 32$ affects the test loss over the course of training. We observe a uniform improvement in test loss across epochs when averaging is applied. This is consistent with our theoretical guarantees, which show that logarithmic-sized subsets approximate full group averaging with a uniform error bound that holds for all square-integrable functions, and hence is reflected uniformly over training as the learned function evolves.

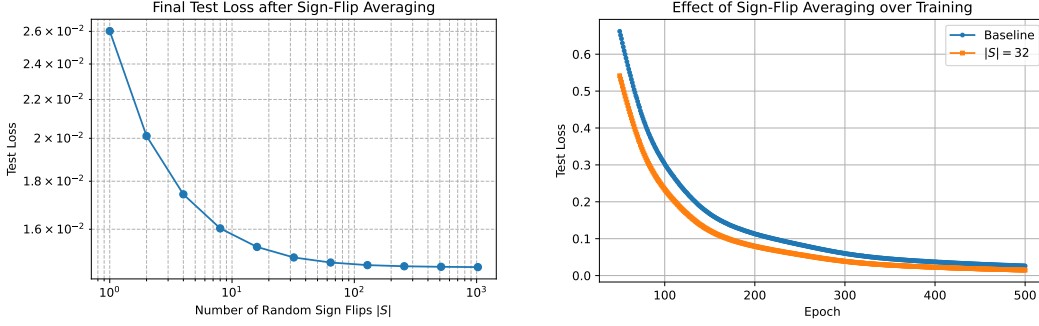

Figure 2: Left: Final test loss when averaging over random subsets $S \subset G$ of increasing size $|S|$. Most of the benefit is achieved already at $|S| = 32$, with only marginal gains beyond that. Right: Test loss over training epochs, with and without averaging using a subset of size $|S| = 32$. The improvement from averaging is observed uniformly over training.

