# OpenReview forum: "Achieving Approximate Symmetry Is Exponentially Easier than Exact Symmetry"
_ICLR.cc/2026/Conference — ICLR 2026 Poster_

### Official Review · Reviewer_9twD · 2025-10-23

**Soundness:** 2
**Presentation:** 2
**Contribution:** 2
**Rating:** 4
**Confidence:** 3

**Summary:**

Summary

This paper investigates the theoretical trade-off between enforcing exact versus approximate symmetry in machine learning models. While exact symmetry is known to act as a strong inductive bias and often improves performance in scientific domains, recent empirical evidence suggests that approximate symmetry can yield greater flexibility and robustness. However, a formal understanding of the cost difference between these two paradigms has been missing.
To address this gap, the authors introduce “averaging complexity,” a new framework that quantifies the computational cost of imposing symmetry through averaging operations. Their main theoretical result establishes an exponential separation: under standard conditions, exact symmetry requires linear averaging complexity, while approximate symmetry can be achieved with only logarithmic complexity.
This finding provides the first formal justification for the practical advantages of approximate symmetry, showing that small relaxations of exact invariance can substantially reduce computational cost without sacrificing desirable structural properties. Beyond this specific result, the proposed analytical tools and complexity framework are likely to be of independent theoretical interest for the broader study of symmetries and inductive biases in machine learning.

**Strengths:**

The paper’s core strength is its clean, theory-driven comparison of exact vs. approximate symmetry, a question that is both timely and genuinely interesting given how often practitioners trade strict equivariance for flexibility. The authors introduce a model-agnostic notion of “averaging complexity” that crisply quantifies the cost of enforcing symmetry via action queries, and they prove a striking exponential separation: exact symmetry needs linear complexity in |G|, whereas approximate symmetry can be achieved with logarithmic complexity. This result not only formalizes widespread empirical intuition—that approximate symmetry is easier and often more robust in practice—but also provides actionable design insight for symmetry-aware learning (e.g., symmetry discovery or semi-supervised settings where small violations are acceptable). Technically, the framework is conceptually simple yet broadly applicable, with proofs that leverage representation theory in a transparent way and yield constructive bounds (e.g., the degree K controlling when exact symmetry becomes linearly expensive). The work thus bridges a real gap between practice and theory, offering a principled rationale for when and why approximate symmetry should be preferred, and supplying tools likely useful beyond the specific results (e.g., for analyzing other symmetry-enforcing pipelines such as group averaging, canonicalization, or data augmentation).

**Weaknesses:**

The paper tackles an interesting and timely question—contrasting exact vs. approximate symmetry—but its technical development feels overly elementary relative to the ambition of the claim. Most arguments rely on standard representation-theoretic decompositions, Fourier analysis on finite groups, and matrix concentration, yielding a clean but fairly direct O(log |G| / ε) upper bound and a linear lower bound via a Vandermonde-style character argument. As a result, the “exponential separation’’—while appealing—rests on a simplified averaging model (x-independent linear post-processing with action queries) and finite groups, leaving unclear how far the conclusions extend to practically important settings (compact Lie groups such as SO(3)/E(3), continuous domains, learned filters, or non pointwise mechanisms). The lower bound for exact symmetry hinges on tensor-power feature lifts and a bound on K via the number of distinct character values; this can be loose, and the paper does not establish matching lower bounds for approximate symmetry or tight constants (or necessity) in the O(\log |G|/\varepsilon) rate. Moreover, the framework abstracts away data- or state-dependent averaging (e.g., frame averaging with x-dependent weights), model-dependent implementations (equivariant layers, steerable kernels), and optimization constraints, so the gap between complexity-in-principle and trainable procedures remains wide. From a learning-theoretic perspective, the results are existential and asymptotic: there are no generalization/error-rate guarantees, no minimax or sample-complexity characterizations tied to the symmetry deficit, and no robustness analysis to noise or misspecification. In short, while the problem is compelling, the paper would benefit from strengthening the main theorems (e.g., tighter—ideally optimal—bounds and explicit constants, extensions beyond finite groups, lower bounds for approximate symmetry, or constructive/algorithmic realizations), thereby elevating the contribution beyond what can be achieved with relatively standard tools.

**Questions:**

1.	Lower Bounds and Tightness: The paper establishes an upper bound of  O(\log |G| / \varepsilon)  for approximate symmetry.
Can the authors characterize whether this rate is tight? In particular, are there matching lower bounds on averaging complexity that depend on the function class F?
2.	Extension to Continuous or Lie Groups: The current framework assumes finite groups. How would the results change for compact or continuous groups (e.g., SO(3), SE(3)) where integration replaces summation? Would the averaging complexity still exhibit a logarithmic versus linear separation under appropriate discretization or measure-theoretic assumptions?
3.	Generalization and Robustness: Since approximate symmetry is shown to be easier to enforce, can this framework be extended to analyze generalization error or robustness to distributional shifts?
For instance, can one derive bounds on sample complexity or Rademacher complexity as a function of the averaging complexity or \varepsilon?
4.	Practical Implications for Optimization: In practice, symmetry enforcement interacts with gradient-based optimization.
How does probabilistic averaging (with only a few sampled group elements) affect gradient variance, bias, and convergence stability during training?
5.	Function Space Dependence: The analysis is primarily framed for continuous functions in L^2(X). Could the authors extend their results to other function spaces—such as Sobolev, Barron, or RKHS spaces—and derive approximation rates depending on smoothness or polynomial degree?
6.	Alternative Norms and Metrics: The results rely on L^2-based definitions of approximate symmetry.
Would the same scaling laws hold under supremum norms or other distances relevant to robust learning or adversarial settings?
7.	Adaptive or Data-Dependent Averaging: If the averaging scheme were made adaptive or data-dependent (e.g., learned during training), could one further reduce the \log |G| scaling? Are there theoretical barriers preventing such improvements?

---

> ### Author Response · Authors · 2025-12-04
>
> Thanks for the review. Here is our response:
>
>
> > ... but its technical development feels overly elementary relative to the ambition of the claim. Most arguments rely on standard representation-theoretic decompositions, Fourier analysis on finite groups, and matrix concentration, yielding a clean but fairly direct O(log |G| / ε) upper bound and a linear lower bound via a Vandermonde-style character argument.
>
> **Answer:** We politely disagree with the respected reviewer. The bounds we derive for approximate symmetry and exact symmetry, and the exponential separation, are completely new to the geometric ML community. It is neither "elementary" nor "standard," and nothing in our proof is a "direct" conclusion. We formulated this problem for the first time in the community and derived highly non-trivial results.
>
>
>
> > ... leaving unclear how far the conclusions extend to practically important settings (compact Lie groups such as SO(3)/E(3), continuous domains, learned filters, or non pointwise mechanisms).
>
> > Question 2: Extension to Continuous or Lie Groups: The current framework assumes finite groups. How would the results change for compact or continuous groups (e.g., SO(3), SE(3)) where integration replaces summation? Would the averaging complexity still exhibit a logarithmic versus linear separation under appropriate discretization or measure-theoretic assumptions?
>
> **Answer:** The application to infinite groups makes no sense since the infinite group case, for full group averaging, involves an integration and thus cannot ever be approximated via a finite sum. The "logarithm" does not provide any meaning here since $\log \infty = \infty$. Finding appropriate formulation/results for infinite groups is an interesting future direction that we already mentioned in the conclusion of our paper. We believe the case of finite groups (e.g., permutations) has its own independent interest in the community, and that its unapplicability to infinite groups is not an issue.
>
> Moreover, we are not sure what you mean by "continuous" domain since we already covered this in the paper. The application to "learned filters" is irrelevant and out of scope of the paper, and it is not clear what the reviewer means by "non pointwise mechanisms"  in their comment.
>
>
> > ... the paper does not establish matching lower bounds for approximate symmetry or tight constants (or necessity) in the O(\log |G|/\varepsilon) rate.
>
> > Question 1: Lower Bounds and Tightness: The paper establishes an upper bound of O(\log |G| / \varepsilon) for approximate symmetry. Can the authors characterize whether this rate is tight? In particular, are there matching lower bounds on averaging complexity that depend on the function class F?
>
> **Answer:** We provided this and proved the optimality (i.e., lower bounds) in the new version of the paper, see Appendix E in the new version.
>
> > ...  the framework abstracts away data- or state-dependent averaging (e.g., frame averaging with x-dependent weights), model-dependent implementations (equivariant layers, steerable kernels), and optimization constraints, so the gap between complexity-in-principle and trainable procedures remains wide.
>
> > Question 4: Practical Implications for Optimization: In practice, symmetry enforcement interacts with gradient-based optimization. How does probabilistic averaging (with only a few sampled group elements) affect gradient variance, bias, and convergence stability during training?
>
> **Answer:**  The main objective of the paper is to study approximate symmetry when it comes to group averaging, where we formulated it as the so-called action queries.
>
> Frame-averaging has a number of issues that our method does not have: it is sometimes discontinuous (while ours is always smooth), and it is hard to find (while ours is easy by random sampling). Frame averaging is also exactly symmetric and thus irrelevant to the study of approximate symmetry. The study of "equivariant layers," "steerable kernels," "optimization," and training is also irrelevant. The paper studies group averaging, which has its own importance and is a subject of independent study in geometric ML.
>
> That being said, we believe that the study of how we can do optimization under symmetries using the approximate averaging is an interesting future direction for this work.

---

> > ### Author Response · Authors · 2025-12-04
> >
> > > From a learning-theoretic perspective, the results are existential and asymptotic: there are no generalization/error-rate guarantees, no minimax or sample-complexity characterizations tied to the symmetry deficit
> >
> >
> > > Question 3: Generalization and Robustness: Since approximate symmetry is shown to be easier to enforce, can this framework be extended to analyze generalization error or robustness to distributional shifts? For instance, can one derive bounds on sample complexity or Rademacher complexity as a function of the averaging complexity or \varepsilon?
> >
> > **Answer:**  The reviewer unfortunately didn't check the appendix of the paper. We already derived sample complexity of generalization bounds for approximate symmetry (using the proposed method) and compared it with exact symmetry in Appendix A.8 in the new version (we emphasize it was available in the reviewed version of the paper, and we didn't change it over the rebuttal)
> >
> > Moreover, the study of "robustness to distribution shifts" is out of scope of our paper, and it is irrelevant.
> >
> >
> > > Question 5: Function Space Dependence: The analysis is primarily framed for continuous functions in L^2(X). Could the authors extend their results to other function spaces—such as Sobolev, Barron, or RKHS spaces—and derive approximation rates depending on smoothness or polynomial degree?
> >
> >
> > **Answer:**  RKHs, Sobolev, and Barron spaces are subsets of L^2, thus our results are already applicable to such settings! We already cover these!
> >
> >
> > > Question 6: Alternative Norms and Metrics: The results rely on L^2-based definitions of approximate symmetry. Would the same scaling laws hold under supremum norms or other distances relevant to robust learning or adversarial settings?
> >
> > **Answer:** We provide a new detailed discussion section in Appendix F on why we chose the L^2 norm. In short, the L^2 norm allows for achieving a universal bound over the function class, does not involve covering arguments and unknown big constants in the upper bound, and it is directly related to the sample complexity and generalization bounds (Appendix A.8). Please refer to Appendix F for more discussion.
> >
> >
> > > Question 6: Adaptive or Data-Dependent Averaging: If the averaging scheme were made adaptive or data-dependent (e.g., learned during training), could one further reduce the \log |G| scaling? Are there theoretical barriers preventing such improvements?
> >
> > **Answer:** Even with just summing over one element, via canonicalization, one can achieve exact symmetry. However, these data-dependent methods are reported in the community to be discontinuous (Dym et. al. 2024), and computationally hard (i.e., graph isomorphism), and while approximate symmetry via averaging is smooth and easy. They are two different methods with their own pros/cons.

---

### Official Review · Reviewer_gGF9 · 2025-10-30

**Soundness:** 3
**Presentation:** 2
**Contribution:** 2
**Rating:** 4
**Confidence:** 4

**Summary:**

The authors study group averaging procedures for achieving group equivariance. Given a finite group G, a group averaging scheme is a function $\omega: G\to \R$. The group averaging scheme induces a linear averaging operator $[\mathbb{E}_\omega(f)](x) = \sum_{g\in G}\omega(g) f(g^{-1}x)$. The 'trivial choice' of $\omega(g)=1/\abs{G}$ for all $g\in G$ yields an operator for which $\mathbb{E}_\omega f$ always is invariant to the group action.

The authors define the size of $\omega$ as the number of non-zero function values $\omega(g)$. They then pose the question: How large must $\omega$ be in order to achieve perfect symmetrization for a given function class $\mathcal{F}$, and how does this compare to approximate symmetry? By approximate symmetrization, they thereby mean schemes that reduce the equivariance error by a factor $\varepsilon$, as measured in $L^2$-norm.

The main results are twofold: First, they show that as long as the function class $\mathcal{F}$ as a representation of the group contains every irrep of the group, the only scheme that achieves perfect symmetry is the trivial choice -- in particular, the size of $\omega$ necessarily is equal to $\vert{G}\vert$. As a contrast, one can use random constructions to construct $\omega$ of size $\log (\vert G \vert)/\varepsilon$

**Strengths:**

The results of the paper are (mostly, see below for details) correct. The connections between Fourier analysis for finite groups and sampling complexity for achieving invariance through averaging that the authors have discovered are interesting, and they do potentially open up for more research.

**Weaknesses:**

1. The authors make quite far-reaching claims about their results: They state that their results explain why data rarely is exactly equivariant, and why models exploit approximate symmetries more easily. One has to ask if this is really what they show? What they say is that the only way to achieve exact symmetry through averaging, which is by far not the only way of doing so, is to take an uniform average over the whole group, and that this will need many samples given the function class is, from a representation theory perspective, is complicated enough. The existence of steerable neural networks shows that enforcing symmetry is very possible, and implementing them is arguably not fundamentally harder than implementing any other neural network.

2. Assumption 9 is a lot stronger than the authors make it out to be. For one, it implies assumption 8: If there is a non-trivial group element $g$ for which $gx=x$ for all $x$, there surely cannot exist a function $f$ (in or not in $\mathcal{F}$) with $f(gx)\neq f(x)$. In fact, by essentially the same argument, it implies that each element $x$ in the domain has a non-trivial stabiliser $\{ h \, \vert \, hx=x\}$. This rules out e.g. the canonical action of the permutation group on $\mathbb{R}^d$. Since the authors ultimately only need that the action of the group on $\mathcal{F}$ is faithful, I am unsure why they make this assumption.

3. Definition 10 is as far as I am concerned not the standard definition of a tensor power of function spaces: The elements of the $\mathcal{F} \otimes \mathcal{F}$, for instance, are rather functions on the space $\mathcal{X}\times \mathcal{X}$, with $(f\otimes g)(x,y) = f(x)g(y)$. I think that this is not just a question of conventions -- note that in order for the representation theoretic machinery to work (in particular that the characters of representation on $\mathcal{F}^{\otimes k}$ become $k$:th powers of the characters of the representation on $\mathcal{F}$), the canonical inner product $\langle x \otimes y, z \otimes w\rangle = \langle x,z \rangle \langle y,w \rangle$ needs to be used. This is not what happens when we're forming polynomials from function -- the scalar product is still the $L^2(\mathcal{X})$-scalar product!

4. There are some mistakes in the reference list. The title of [Bourgain, Gamburd] is missing an $\mathrm{SL}_2(\mathbb{F}_p)$,  [Huang, Levie,Villar] and [Tahmasebi,Jegelka;2023] do not have journals. Furthermore, I can't find the paper "A differentiable metric for discovering groups and unitary representations" by Dongsung Huh, ICLR 2025. Has the paper changed name at some point?

5. Some parts of the text in the appendix do not flow very well. There are e.g. parts of the texts that are repeated verbatim in several proofs.

**Questions:**

1. Can the authors provide some more arguments why $\mathrm{AC^{ex}}$, $\mathrm{AC^{wk}}$ and $\mathrm{AC^{st}}$ is a good way of measuring the hardness of enforcing symmetries?

2. Is there a specific reason why the authors do not just assume that the action of $G$ on $\mathcal{F}$ is faithful directly?

3. Can the apparent problem that the definition of the tensor product doesn't seem to be correct be fixed?

4. Is the result that $\mathrm{K}$ can be chosen as $\vert \lbrace \ \mathrm{Tr}(\rho(g)) \ \vert \ g \in G \rbrace \vert$ known or new?

5. In the proof of Theorem 13, can the authors point to the concrete matrix concentration inequality they apply in row 1453?

---

> ### Author Response · Authors · 2025-12-04
>
> Thanks for your review. Here is our response to the comments:
>
>
>
>
>
>
>
> > Weakness 4
>
>
> **Answer:** We corrected the typos in the new version. Also, the paper that the reviewer could not find is indeed an ICLR paper 2025 with a new name, "Discovering Group Structures via Unitary Representation Learning," and they changed the name of this paper after their paper was accepted at ICLR. We already corrected this in the paper, and we wanted to mention that the paper we cited in the previous version indeed has been the previous version of this ICLR 2025 paper on OpenReview, that we didn't update its name (while we already had the name of the venue). Thanks for pointing out this typo.
>
>
>  > Question 1: Can the authors provide some more arguments why .. and ..  is a good way of measuring the hardness of enforcing symmetries?
>
> **Answer:** We wanted to see how much group averaging can be compressed, as a notion of complexity. Intuitively, the more we relax the symmetry condition, the more we can compress group averaging, and that's the whole intuition.
>
>
>
>  > Question 2: Is there a specific reason why the authors do not just assume that the action of .. on .. is faithful directly?
>
> > Weakness 2
>
> **Answer:** Thanks for pointing this out. This is correct, we followed your suggestion and just mentioned that the action on the function space is faithful, as the main assumption (so the two assumptions are now combined together, and we just have Assumption 9 in the revised version).
>
>
>  > Question 3: Can the apparent problem that the definition of the tensor product doesn't seem to be correct be fixed?
>
> > Weakness 3
> >
> **Answer:** Thanks for your comment. Yes, in the first version, we mistakenly used the term tensor product and its notation for deriving the result, but following up on your concern, we realized that the appropriate term for what we are using is "symmetric tensor powers," and thus we changed all notations and adapted results to this case in the new revised version. Nothing serious has changed, and the message and results remain unchanged. We just changed the notations, definition, and adapted the results to the correct notion of tensor powers.
>
>
>
>  > Question 4: Is the result that .. can be chosen as ... known or new?
>
> **Answer:** The bound is used from a known fact (recently derived) in pure mathematics (algebra). We provide citations to relevant references in Claim 22 in the revised version. However, we are the first ones to use such methods to obtain bounds for approximate group averaging, to the best of our knowledge.
>
>
>
>
> > Question 5: In the proof of Theorem 13, can the authors point to the concrete matrix concentration inequality they apply in row 1453?
>
>
> **Answer:** We provided concrete citations to the theorem we are using (Line 1475 in the new version).

---

### Official Review · Reviewer_SnX1 · 2025-10-31

**Soundness:** 3
**Presentation:** 3
**Contribution:** 3
**Rating:** 6
**Confidence:** 3

**Summary:**

This paper investigates the theoretical complexity of enforcing exact versus approximate symmetry in machine learning models. The authors introduce a novel framework called "averaging complexity" to quantify the cost of symmetry enforcement through action queries. Their main contribution is proving an exponential separation: exact symmetry requires linear averaging complexity in the group size, while approximate symmetry needs only logarithmic complexity. The paper uses representation theory tools to establish these bounds, providing the first theoretical justification for why approximate symmetry is often preferred in practice.

**Strengths:**

Originality and Theoretical Contribution

The paper addresses a significant gap in the literature by providing the first direct theoretical comparison between exact and approximate symmetry enforcement. The exponential separation result (linear versus logarithmic complexity) is novel and formally justifies intuitions from empirical work. The averaging complexity framework itself is a creative abstraction that could enable future theoretical analyses in geometric machine learning.

Mathematical Rigor and Technical Quality

The proofs leverage sophisticated representation theory, including character theory, Fourier analysis on groups, and tensor product decompositions. The constructive proof of Theorem 11 provides explicit bounds on the threshold K where exact symmetry requires full group averaging. The probabilistic construction in Theorem 13 using matrix concentration inequalities is elegant and yields tight logarithmic bounds.

Practical Relevance

The work connects to important practical considerations in geometric machine learning, including symmetry discovery, robustness to distributional shift, and semi-supervised learning. The theoretical insights help explain why approximate symmetry methods have been successful across various applications from medical imaging to molecular modeling.

**Weaknesses:**

Limited Scope to Finite Groups

The entire framework is restricted to finite groups, which significantly limits applicability. Many important symmetries in machine learning involve continuous groups like SO(3) for rotations or SE(3) for molecular systems. The authors acknowledge this limitation but provide no path forward. The technical machinery (character theory, Fourier analysis on finite groups) does not obviously extend to compact Lie groups.

While averaging complexity is mathematically elegant, its relationship to practical computational cost or sample complexity is unclear. Action queries are a theoretical construct, but real systems face different bottlenecks (gradient computation, memory, data requirements). The paper would benefit from discussing how averaging complexity relates to these practical concerns.

The paper is purely theoretical with no experiments whatsoever. Even simple synthetic experiments demonstrating the exponential separation on toy problems would strengthen the work. Without empirical validation, it is difficult to assess whether the constants hidden in the big-O notation matter in practice, or whether the assumptions (faithful action, orbit separability) hold for realistic function classes.

Assumptions 8 and 9 (faithful group action and orbit separability) are needed for Theorem 11 but not for Theorem 13. The paper does not thoroughly discuss when these assumptions hold in practice or provide counterexamples when they fail. For tensor powers reaching degree K, the dimension of the function class grows exponentially, which may be impractical for large groups.

The paper does not compare averaging complexity to other complexity measures in learning theory such as VC dimension, Rademacher complexity, or sample complexity. Understanding these relationships would better situate the contribution within the broader theoretical landscape.

**Questions:**

Can the authors provide any insight into whether similar results might hold for compact Lie groups? What are the fundamental technical barriers? Would one need to replace action queries with a different notion of complexity?

How should practitioners use these results? Does the logarithmic bound for approximate symmetry suggest specific algorithmic approaches? Can the probabilistic construction in Theorem 13 be turned into a practical sampling strategy?

The authors mention constants differ by at most a factor of four between weak and strong approximate symmetry. What are the actual constants? Are they small enough that the logarithmic bound is meaningful for realistic group sizes?

Are the bounds tight? Is there a matching lower bound showing that logarithmic complexity is necessary for approximate symmetry? Can the authors provide examples where the trivial K ≤ |G| bound is achieved versus cases where K is much smaller?

Can the authors provide concrete examples of important function classes or group actions where Assumptions 8 and 9 fail? What happens to the results in these cases?

---

> ### Author Response · Authors · 2025-12-04
>
> Thanks for your review. Here is our response to the comments:
>
>  > The entire framework is restricted to finite groups, which significantly limits applicability. Many important symmetries in machine learning involve continuous groups like SO(3) for rotations or SE(3) for molecular systems. The authors acknowledge this limitation but provide no path forward. The technical machinery (character theory, Fourier analysis on finite groups) does not obviously extend to compact Lie groups.
>
>
> > Can the authors provide any insight into whether similar results might hold for compact Lie groups? What are the fundamental technical barriers? Would one need to replace action queries with a different notion of complexity?
>
>
> **Answer:**  Thanks for your question. Since group averaging for infinite groups is a continuous integral, there is no finite sum that uniformly approximates it in L^2 space (in other words, it has infinitely many irreps). One can, of course, consider an extra assumption that only finitely many irreps appear in the representation of the function space, and that allows one to directly conclude a fairly similar result, but it involves some extra conditions that are beyond the scope of this paper and need a fairly independent study. We mention this in the conclusion section of our paper.
>
>
>
>  > While averaging complexity is mathematically elegant, its relationship to practical computational cost or sample complexity is unclear. Action queries are a theoretical construct, but real systems face different bottlenecks (gradient computation, memory, data requirements). The paper would benefit from discussing how averaging complexity relates to these practical concerns.
>
>
> >  How should practitioners use these results? Does the logarithmic bound for approximate symmetry suggest specific algorithmic approaches? Can the probabilistic construction in Theorem 13 be turned into a practical sampling strategy?
>
>
>  > The paper is purely theoretical with no experiments whatsoever. Even simple synthetic experiments demonstrating the exponential separation on toy problems would strengthen the work. Without empirical validation, it is difficult to assess whether the constants hidden in the big-O notation matter in practice, or whether the assumptions (faithful action, orbit separability) hold for realistic function classes.
>
>
>
> **Answer:** Thanks for your comment. The averaging complexity is nothing but trying to approximate full group averaging with an average over a fairly small subset of the group. In our result, we prove that a uniform approximation in L^2 distance is possible if the size of the subset is just logarithmic in the group size, and one can use randomness (i.e., no computational barrier) to find such a set. The result applies to neural networks as well, as our main result for approximate symmetries does not have any condition that excludes neural networks.
>
> Moreover, in the new version, we added experiments that verify the theoretical findings, and we used neural networks as our setting. Please check them in Appendix G.
>
>
>
>  > Assumptions 8 and 9 (faithful group action and orbit separability) are needed for Theorem 11 but not for Theorem 13. The paper does not thoroughly discuss when these assumptions hold in practice or provide counterexamples when they fail. For tensor powers reaching degree K, the dimension of the function class grows exponentially, which may be impractical for large groups.
>
>
> > Can the authors provide concrete examples of important function classes or group actions where Assumptions 8 and 9 fail? What happens to the results in these cases?
>
>
> **Answer:**  Following the suggestion of another reviewer, we combine these two into the new Assumption 9 in the revised version of the paper. The assumption essentially means that the group action on the function space is faithful, which means that any group element acts non-trivially on the function space. It is fairly easy to verify for any model (including neural networks), and it is barely possible to find any meaningful practical case where checking this assumption (and observing that it is violated) is not easy.
>
>
>
>
> > The paper does not compare averaging complexity to other complexity measures in learning theory such as VC dimension, Rademacher complexity, or sample complexity. Understanding these relationships would better situate the contribution within the broader theoretical landscape.
>
> **Answer:** Other complexity measures that the reviewer mentioned are all related to generalization error, while our theory concerns approximation error, and this is a fundamentally different quantity, and it is not comparable with them. That being said, we already have generalization and sample complexity bounds in the paper (Appendix A.8), and the relation of our theory to generalization is already explained in detail in the paper.

---

> > ### Author Response · Authors · 2025-12-04
> >
> > > The authors mention constants differ by at most a factor of four between weak and strong approximate symmetry. What are the actual constants? Are they small enough that the logarithmic bound is meaningful for realistic group sizes?
> >
> > **Answer:** We reported the actual constants in the new version of the paper. These are $2.67$ and $10.67$, which are weak and strong cases, respectively, and thus they are meaningful.
> >
> >
> >
> > > Are the bounds tight? Is there a matching lower bound showing that logarithmic complexity is necessary for approximate symmetry? Can the authors provide examples where the trivial K ≤ |G| bound is achieved versus cases where K is much smaller?
> >
> >
> >
> > **Answer:** We added a new section to the paper (Appendix E) and proved that a logarithmic lower bound as well. So that result is now having a matching lower bound in the paper. Also, having better bounds on K is a fairly complicated algebraic problem and is an open area of research in mathematics. That being said, for neural networks, we believe that since they are universal, such an assumption on K is already satisfied by them. This intuition follows from the universality.

---

### Official Review · Reviewer_hsGK · 2025-11-01

**Soundness:** 3
**Presentation:** 2
**Contribution:** 3
**Rating:** 6
**Confidence:** 2

**Summary:**

This paper presents the first formal theory comparing the enforcement difficulty of exact and approximate symmetry in machine learning models. By introducing the notion of averaging complexity, the authors prove an exponential separation: exact symmetry requires linear complexity in the group size, while approximate symmetry needs only logarithmic complexity. Using representation theory and Fourier analysis, the work explains why approximate equivariance is both more flexible and computationally efficient, offering a principled foundation for its empirical success in geometric deep learning.

**Strengths:**

The paper’s main theorem establishes an exponential separation between exact and approximate symmetry, supported by rigorous analysis using representation theory, Fourier analysis, and probabilistic tools. The writing is clear and well-structured, effectively connecting the theoretical insights to practical implications for generalization and robustness in symmetry-based learning.

**Weaknesses:**

1. The paper lacks empirical or numerical evidence to support its theoretical claims. Including simple synthetic experiments, such as verifying the predicted scaling laws of averaging complexity in finite groups, would make the results more concrete and help readers connect the abstract theory to observed model behavior.
2. While the proofs are rigorous, the intuition behind key representation-theoretic arguments, such as the role of tensor powers and character separability in explaining the linear versus logarithmic complexity gap, could be better articulated. Providing illustrative examples of other groups, such as cyclic or dihedral groups, would further improve clarity and highlight the generality of the framework.

**Questions:**

1. The paper focuses on finite groups throughout the analysis. How challenging would it be to extend the notion of averaging complexity to continuous groups, such as $\mathrm{SO}(3)$ or $\mathrm{SE}(3)$? Do the authors expect a similar exponential separation between exact and approximate symmetry to hold in that case, or are there fundamental obstacles?

2. The exponential separation result relies on representation-theoretic properties and tensor-power constructions. Could the authors provide additional intuition about the role of the number of distinct character values $K = \\{\mathrm{Tr}(\rho(g)) : g \in G\\}$? In particular, how should we interpret this parameter in concrete machine learning models?

3. How does averaging complexity relate to empirical training complexity in neural networks that are approximately equivariant? For example, can this framework provide any guidance on the number of samples or architectural constraints required to achieve a desired level of approximate symmetry in practice?

---

> ### Author Response · Authors · 2025-12-04
>
> Thanks for your review. Here is our response to the comments:
>
>  > The paper lacks empirical or numerical evidence to support its theoretical claims. Including simple synthetic experiments, such as verifying the predicted scaling laws of averaging complexity in finite groups, would make the results more concrete and help readers connect the abstract theory to observed model behavior.
>
>
> **Answer:** Thanks for your suggestion. We included a set of experiments in Appendix G in the new version of our paper, which verifies the scaling laws, uniformity, and logarithmic size effectiveness of approximate group averaging.
>
>
>
>
> > Question 1: The paper focuses on finite groups throughout the analysis. How challenging would it be to extend the notion of averaging complexity to continuous groups, such as SO(3) or SE(3)? Do the authors expect a similar exponential separation between exact and approximate symmetry to hold in that case, or are there fundamental obstacles?
>
> **Answer:** Thanks for your question. Since group averaging for infinite groups is a continuous integral, there is no finite sum that uniformly approximates it in L^2 space (in other words, it has infinite irreps). One can, of course, consider an extra assumption that only finitely many irreps appear in the representation of the function space, and that allows one to directly conclude a fairly similar result, but it involves some extra conditions that are beyond the scope of this paper and need a fairly independent study. We mention this in the conclusion section of our paper.
>
>
> > Question 2: The exponential separation result relies on representation-theoretic properties and tensor-power constructions. Could the authors provide additional intuition about the role of the number of distinct character values K? In particular, how should we interpret this parameter in concrete machine learning models?
>
> **Answer:** In Example 12, we provided an example of computing the bound.
>
> In general, since neural networks are rich in the number of parameters and uniform approximators, we believe that they satisfy the condition on K, and thus they need full group averaging. Thus, for deep learning, we interpret the condition on K as being less strict and fairly easy to satisfy.
>
>
> > Question 3: How does averaging complexity relate to empirical training complexity in neural networks that are approximately equivariant? For example, can this framework provide any guidance on the number of samples or architectural constraints required to achieve a desired level of approximate symmetry in practice?
>
>
> **Answer:** Yes, the architecture directly applies to neural networks, as the only assumption is being in L^2 (i.e., square-integrability) that holds for neural networks for appropriate distributions or compact data domains. The logarithmic bound does not depend on any tensor product or polynomial feature assumption; it is general.

---

### Official Review · Reviewer_MdPu · 2025-11-05

**Soundness:** 3
**Presentation:** 4
**Contribution:** 3
**Rating:** 8
**Confidence:** 4

**Summary:**

This paper examines the question of the efficiency of enforcing exact equivariant compared to approximate invariance from a theoretical point of view. A framework is developed to precisely characterize the notions of efficiency and approximate invariance in this context. The authors show that, under general assumptions, enforcing exact invariance is exponentially harder than enforcing approximate invariance.

**Strengths:**

- The paper tackles important and timely questions in geometric deep learning, namely what are the benefits and cost of exact equivariance vs approximate equivariance.
- The results provided are nontrivial and sound as far as I could check. The formalism could probe useful to prove other results.
- The paper is well written and clear

**Weaknesses:**

- The assumption of the function class being a finite-dimensional vector space seems quite strong and potentially restrictive. Can the authors comment on that in the paper?
- Consider including experimental verification of the claim of logarithmic scaling of averaging complexity for approximate invariance (could be in the appendix)
- The definition of symmetry for a function coincidences with what is more commonly referred to as invariance, I suggest using that term instead for clarity
- It seems like the analysis is restricted to invariance and not equivariance. However, this is not clear from the start of the paper. Could the authors clarify that?

**Questions:**

- I feel like an additional step that could make the story more complete is a discussion of the potential generalization benefits of approximate invariance. I don't know if such results already exist, maybe a connection to Elesedy and Zaidi 2021 could be drawn.

---

> ### Author Response · Authors · 2025-12-04
>
> Thanks for your review and support of our manuscript. Here is our response to the comments:
>
>  > The assumption of the function class being a finite-dimensional vector space seems quite strong and potentially restrictive. Can the authors comment on that in the paper?
>
>
> **Answer:** Thanks for your comment. Our result applies to ANY finite-dimensional function class, thus even if you consider an infinite-dimensional function class (subset of L^2), as far as the group representation decomposes into irreps, you can achieve the same result. The result is essentially for the whole L^2 space, under the assumption that the group representation decomposes into irreps. This assumption is a convention in representation theory, and unless we consider pathological cases, it applies to all practical cases of our problem.
>
> So essentially, you can obtain the result for infinite-dimensional function spaces for free.
>
>
>
>
>  > Consider including experimental verification of the claim of logarithmic scaling of averaging complexity for approximate invariance (could be in the appendix)
>
> **Answer:** In the new version, in Appendix G, we provided a new set of experiments that provides verification of logarithmic scaling and the uniformity of the approximate symmetry error bound in our result. Please check Appendix G in the new version.
>
>
>
>  > The definition of symmetry for a function coincidences with what is more commonly referred to as invariance, I suggest using that term instead for clarity
>
>
> **Answer:** Thanks for your suggestion, we will follow this.
>
>
>  > It seems like the analysis is restricted to invariance and not equivariance. However, this is not clear from the start of the paper. Could the authors clarify that?
>
>
> **Answer:** Our theory is also valid for equivariance, and we provide an explanation of how we do this in Appendix A.4 in the new version.
>
>
>  > I feel like an additional step that could make the story more complete is a discussion of the potential generalization benefits of approximate invariance. I don't know if such results already exist, maybe a connection to Elesedy and Zaidi 2021 could be drawn.
>
>
> **Answer:** Thanks for your comment, we already have the generalization and sample complexity bounds for approximate invariance in Appendix A.8 in the paper. We already provided all the related discussion in the paper, but in the appendix. In the new version, we provide appropriate pointers to this in the main body of the paper as well.

---

### Meta-Review · Area_Chair_fP8H · 2026-01-06

**Summary:**

1.  *MdPU* has concerns about the strength of the finite dimensional assumption.  *SnX1* notes the paper does not thoroughly discuss the necessity of various assumptions. *gGF9* notes that not all the assumptions are necessary and some imply others.
2. *hsGK* notes some empirical experimental support would strengthen the paper.  *SnX1* notes the lack of experiments and the lack of insight this gives into practical computational complexity.
3. *hsGK* suggests adding examples to make the theory easier to understand.
4. Some reviewers such as *SnX1* noted or asked about the fact the methods do not extend beyond finite groups.
5. *gGF9* thinks the authors overclaim the impact of their results.
6. *gGF9* pointed out the authors definition of tensor product is wrong

**Reviewer Concerns:**

1. The authors note the f.d. assumption is fairly standard and not particularly limiting in practical cases.   They also revise and explain their other assumptions.
2. The authors added experiments and further explain the practicality of their estimates.  They give precise constants for their bounds.
3. This is not addressed, but is not a serious weakness.  The authors do point to examples already included.
4. The authors consider infinite groups beyond the scope of the paper.
5. This is unaddressed.
6. The authors accept the criticism.  They note it should have been called symmetric tensor powers and hold their conclusions are correct.  In this case, further discussion would have been helpful

**Reviewer Scores:**

- *MdPu* would have kept 8
- *hsGK*may have increased from 6 to 8
- *SnX1* there is a decent chance of increase from 6 to 8
- *gGF9* if they accepted the change to symmetric tensor powers may have increased from 4 to 6
- *9twD* gave a 4.  The review feels AI generated and the strengths and weaknesses are very vague.  The authors nonetheless replied. I am  discounting this review.

---

### Decision · Program_Chairs · 2026-01-26

Accept (Poster)